# Co-translational protein targeting facilitates centrosomal recruitment of PCNT during centrosome maturation in vertebrates

Guadalupe Sepulveda[1†], Mark Antkowiak[1†], Ingrid Brust-Mascher[2†], Karan Mahe[1], Tingyoung Ou[1], Noemi M Castro[1], Lana N Christensen[1], Lee Cheung[1], Xueer Jiang[1], Daniel Yoon[1], Bo Huang[3,4,5], Li-En Jao[1]*

[1]Department of Cell Biology and Human Anatomy, University of California, Davis School of Medicine, Davis, United States; [2]Department of Anatomy, Physiology and Cell Biology, University of California, Davis School of Veterinary Medicine, Davis, United States; [3]Department of Pharmaceutical Chemistry, University of California, San Francisco, San Francisco, United States; [4]Department of Biochemistry and Biophysics, University of California, San Francisco, San Francisco, United States; [5]Chan Zuckerberg Biohub, San Francisco, United States

*For correspondence:
ljao@ucdavis.edu

[†]These authors contributed equally to this work

Competing interests: The authors declare that no competing interests exist.

**Abstract** As microtubule-organizing centers of animal cells, centrosomes guide the formation of the bipolar spindle that segregates chromosomes during mitosis. At mitosis onset, centrosomes maximize microtubule-organizing activity by rapidly expanding the pericentriolar material (PCM). This process is in part driven by the large PCM protein pericentrin (PCNT), as its level increases at the PCM and helps recruit additional PCM components. However, the mechanism underlying the timely centrosomal enrichment of PCNT remains unclear. Here, we show that PCNT is delivered co-translationally to centrosomes during early mitosis by cytoplasmic dynein, as evidenced by centrosomal enrichment of *PCNT* mRNA, its translation near centrosomes, and requirement of intact polysomes for *PCNT* mRNA localization. Additionally, the microtubule minus-end regulator, ASPM, is also targeted co-translationally to mitotic spindle poles. Together, these findings suggest that co-translational targeting of cytoplasmic proteins to specific subcellular destinations may be a generalized protein targeting mechanism.
DOI: https://doi.org/10.7554/eLife.34959.001

## Introduction

A centrosome consists of a pair of centrioles embedded in a protein-dense matrix known as the pericentriolar material (PCM). The PCM functions as a major microtubule organizing center in animal cells (*Gould and Borisy, 1977*) as it serves as a platform onto which γ-tubulin ring complexes (γ-TuRCs), the main scaffold mediating microtubule nucleation, are loaded (*Moritz et al., 1995*; *Zheng et al., 1995*).

At the onset of mitosis, centrosomes rapidly expand their PCM. This process, termed centrosome maturation, is essential for proper spindle formation and chromosome segregation (*Woodruff et al., 2014*). Centrosome maturation is initiated by phosphorylation of core PCM components, such as Pericentrin (PCNT) and Centrosomin (Cnn), by mitotic kinases PLK1/Polo and Aurora kinase A (*Conduit et al., 2014a*; *Joukov et al., 2014*; *Kinoshita et al., 2005*; *Lee and Rhee, 2011*). These events then trigger the cooperative assembly of additional PCM scaffold proteins (e.g. PCNT, CEP192/SPD-2, CEP152/Asterless, CEP215/CDK5RAP2/Cnn or SPD-5) into an expanded PCM matrix

**eLife digest** Before a cell divides, it creates a copy of its genetic material (DNA) and evenly distributes it between the new 'daughter' cells with the help of a complex called the mitotic spindle. This complex is made of long cable-like protein chains called microtubules.

To ensure that each daughter cell receives an equal amount of DNA, structures known as centrosomes organize the microtubules during the division process. Centrosomes have two rigid cores, called centrioles, which are surrounded by a matrix of proteins called the pericentriolar material. It is from this material that the microtubules are organized.

The pericentriolar material is a dynamic structure and changes its size by assembling and disassembling its protein components. The larger the pericentriolar material, the more microtubules can form. Before a cell divides, it rapidly expands in a process called centrosome maturation. A protein called pericentrin initiates the maturation by helping to recruit other proteins to the centrosome. Pericentrin molecules are large, and it takes the cell between 10 and 20 minutes to make each one. Nevertheless, the cell can produce and deliver large quantities of pericentrin to the centrosome in a matter of minutes. We do not yet know how this happens.

To investigate this further, Sepulveda, Antkowiak, Brust-Mascher et al. used advanced microscopy to study zebrafish embryos and human cells grown in the laboratory. The results showed that cells build and transport pericentrin at the same time. Cells use messenger RNA molecules as templates to build proteins. These feed into protein factories called ribosomes, which assemble the building blocks in the correct order. Rather than waiting for the pericentrin production to finish, the cell moves the active factories to the centrosome with the help of a molecular motor called dynein. By the time the pericentrin molecules are completely made by ribosomes, they are already at the centrosome, ready to help with the recruitment of other proteins during centrosome maturation.

These findings improve our understanding of centrosome maturation. The next step is to find out how the cell coordinates this process with the recruitment of other proteins to the centrosome. It is also possible that the cell uses similar processes to deliver other proteins to different parts of the cell.

DOI: https://doi.org/10.7554/eLife.34959.002

that encases the centrioles (*Conduit et al., 2014b*; *Hamill et al., 2002*; *Kemp et al., 2004*), culminating in the recruitment of additional γ-TuRCs and tubulin molecules that promote microtubule nucleation and render centrosomes competent for mediating the formation of bipolar spindles and chromosome segregation (*Conduit et al., 2015*; *Gopalakrishnan et al., 2011*; *Woodruff et al., 2014*).

Pericentrin (PCNT) is one of the first core PCM components identified to be required for proper spindle organization (*Doxsey et al., 1994*). Importantly, mutations in *PCNT* have been linked to several human disorders including primordial dwarfism (*Anitha et al., 2009*; *Delaval and Doxsey, 2010*; *Griffith et al., 2008*; *Numata et al., 2009*; *Rauch et al., 2008*). Pericentrin is an unusually large coiled-coil protein (3336 amino acids in human) that forms elongated fibrils with its C-terminus anchored near the centriole wall and the N-terminus extended outwardly and radially across PCM zones in interphase cells (*Lawo et al., 2012*; *Mennella et al., 2012*; *Sonnen et al., 2012*). Recent studies showed that pericentrin plays an evolutionarily conserved role in mitotic PCM expansion and interphase centrosome organization, as loss of pericentrin activity in human, mice, and flies all results in failed recruitment of other PCM components to the centrosome and affects the same set of downstream orthologous proteins in each system (e.g. CEP215 in human, Cep215 in mice, and Cnn in flies) (*Chen et al., 2014*; *Lee and Rhee, 2011*; *Lerit et al., 2015*).

In vertebrates, a key function of PCNT is to initiate centrosome maturation (*Lee and Rhee, 2011*) and serve as a scaffold for the recruitment of other PCM proteins (*Haren et al., 2009*; *Lawo et al., 2012*; *Purohit et al., 1999*; *Zimmerman et al., 2004*). However, the mechanism underlying the timely synthesis and recruitment of a large sum of PCNT proteins to the PCM is as yet unresolved. Given its large size (>3300 amino acids) and the modest rate of translation elongation (~3–10 amino acids per second, *Boström et al., 1986*; *Ingolia et al., 2011*; *Morisaki et al., 2016*; *Pichon et al., 2016*; *Wang et al., 2016*; *Wu et al., 2016*; *Yan et al., 2016*), synthesizing a full-length PCNT protein

would take ~10–20 min to complete after translation initiation. Notably, after the onset of mitosis, the PCM reaches its maximal size immediately before metaphase in ~30 min in human cells (*Gavet and Pines, 2010*; *Lénárt et al., 2007*). Thus, the cell faces a kinetics challenge of synthesizing, transporting, and incorporating multiple large PCM proteins such as PCNT into mitotic centrosomes within this short time frame.

We show here that *pericentrin* mRNA is spatially enriched at the centrosome during mitosis in zebrafish embryos and cultured human cells. In cultured cells, the centrosomal enrichment of *PCNT* mRNA predominantly occurs during early mitosis, concomitantly with the peak of centrosome maturation. We further show that centrosomally localized *PCNT* mRNA undergoes active translation and that acute inhibition of translation compromises the incorporation of PCNT proteins into the centrosome during early mitosis. Moreover, we find that centrosomal localization of *PCNT* mRNA requires intact polysomes, microtubules, and cytoplasmic dynein activity. Taken together, our results support a model in which translating *PCNT* polysomes are being actively transported toward the centrosome during centrosome maturation. We propose that by targeting actively translating polysomes toward centrosomes, the cell can overcome the kinetics challenge of synthesizing, transporting, and incorporating the unusually large PCNT proteins into the centrosome. Lastly, we find that the cell appears to use a similar co-translational targeting mechanism to synthesize and deliver another unusually large protein, the microtubule minus-end regulator, ASPM, to the mitotic spindle poles. Thus, co-translational protein targeting might be a mechanism widely employed by the cell to transport cytoplasmic proteins to specific subcellular compartments and organelles.

## Results

### Zebrafish *pcnt* mRNA is localized to the centrosome in blastula-stage embryos

We found that *pericentrin* (*pcnt*) transcripts were localized to distinct foci in early zebrafish embryos, whereas those of three other core PCM components, *cep152*, *cep192*, and *cep215*, showed a pancellular distribution (*Figure 1A*). This striking *pcnt* mRNA localization was observed using two independent, non-overlapping antisense probes against the 5' or 3' portion of RNA (*Figure 1B*). The specificity of in situ hybridization was further confirmed by the loss of signals in two frameshift maternal-zygotic *pcnt* knockout embryos (MZ*pcnt*$^{tup2}$ and MZ*pcnt*$^{tup5}$) (*Figure 1B* and *Figure 1—figure supplement 1*), where the *pcnt* transcripts were susceptible to nonsense-mediated decay pathway. By co-staining with the centrosome marker γ-tubulin, we demonstrated that zebrafish *pcnt* mRNA is specifically localized to the centrosome (*Figure 1C*).

### Human *PCNT* mRNA is enriched at the centrosome during early mitosis

To test whether centrosomal localization of *pcnt* mRNA is conserved beyond early zebrafish embryos, we examined the localization of human *PCNT* mRNA in cultured HeLa cells using fluorescent in situ hybridization (FISH). Consistent with our observation in zebrafish, human *PCNT* mRNA was also localized to the centrosome (*Figure 2*). Interestingly, this centrosomal enrichment of *PCNT* mRNA was most prominent during early mitosis (i.e. prophase and prometaphase) and declined after prometaphase. The signal specificity was confirmed by two non-overlapping probes against the 5' or 3' portion of the *PCNT* transcript (*Figure 2—figure supplement 1A*). Furthermore, using an alternative FISH method, Stellaris single-molecule FISH (smFISH) against the 5' or 3' portion of the *PCNT* transcript, we observed highly similar centrosomal enrichment of *PCNT* mRNA during early mitosis, with near single-molecule resolution (*Figure 2—figure supplement 1B*). Similar smFISH results were observed in both HeLa and RPE-1 cells (data not shown). Together, these results indicate that *PCNT* mRNA is specifically enriched at the centrosome during early mitosis in cultured human cells. We speculate that the seemingly constant presence of zebrafish *pcnt* mRNA at the centrosome of early blastula-stage embryos is due to the fast cell cycle without gap phases at this stage (~20 min per cycle).

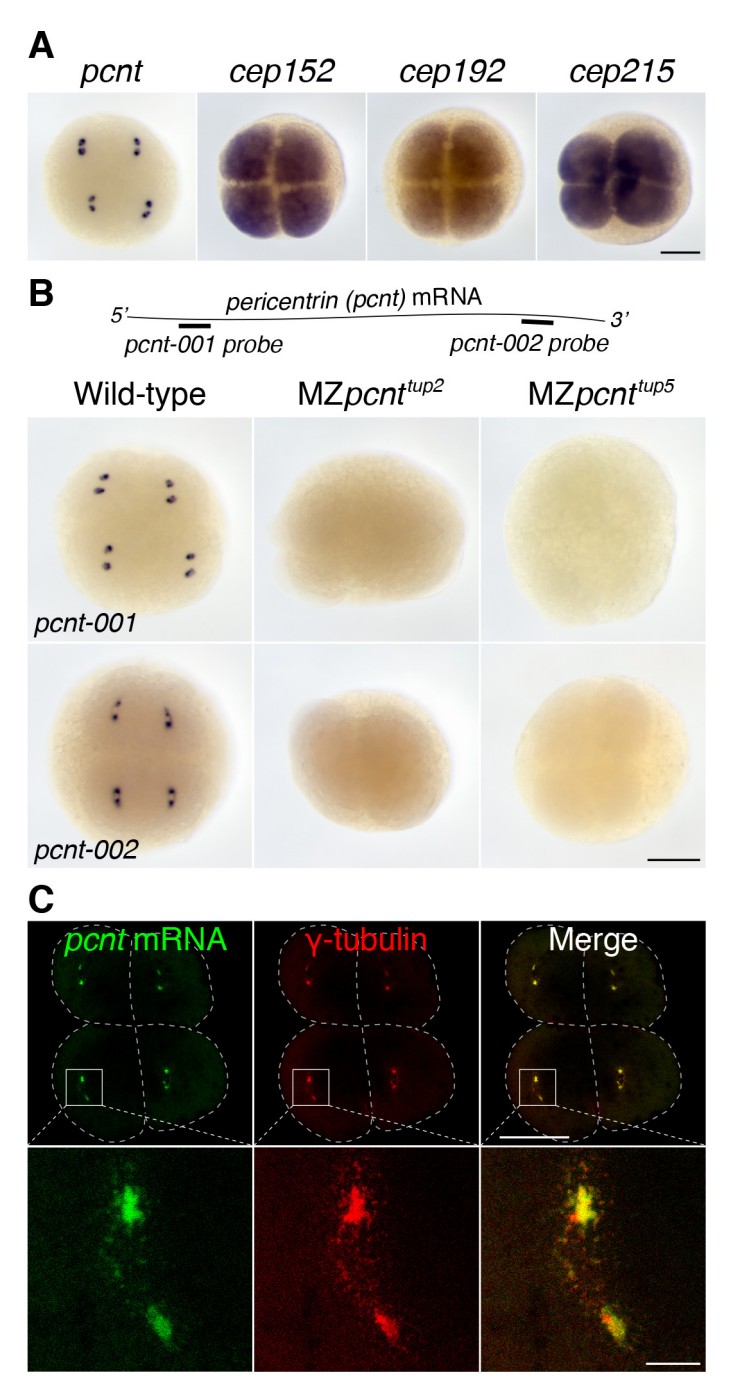

**Figure 1.** *Pericentrin* (*pcnt*) mRNA is localized to centrosomes in early zebrafish embryos. (**A**) RNA in situ hybridization of transcripts of different PCM components in four-cell stage zebrafish embryos. Note that while the mRNA of *cep152*, *cep192*, and *cep215* displayed a pan-cellular distribution, *pcnt* mRNA was concentrated at two distinct foci in each cell. (**B**) RNA in situ hybridization showed similar dot-like patterns of *pcnt* transcripts with two non-overlapping antisense probes. The signals were lost in two maternal-zygotic (MZ) *pcnt* mutants. (**C**) Fluorescent RNA in situ hybridization and anti-γ-tubulin co-staining demonstrated the centrosomal localization of *pcnt* mRNA. n > 300 (*pcnt-001* probe), n > 100 (*pcnt-002* probe), n > 50 (*cep152, cep192, or cep215* probe); all the embryos showed the same RNA distribution patterns as shown in the representative images. More than 100 MZ*pcnt*^*tup2* or MZ*pcnt*^*tup5* embryos were examined; none of them showed visible *pcnt* RNA in situ signals. Embryos were examined between 2- and 16-cell stages with representative four-cell stage embryos shown. Dashed lines delineate the cell boundaries. Scale bars: 200 μm or 25 μm (inset in **C**).

*Figure 1 continued on next page*

*Figure 1 continued*

DOI: https://doi.org/10.7554/eLife.34959.003

The following figure supplement is available for figure 1:

**Figure supplement 1.** Sequences of two Cas9-induced frameshift mutations (alleles *pcnt^{tup2}* and *pcnt^{tup5}*) in the zebrafish *pcnt* gene.

DOI: https://doi.org/10.7554/eLife.34959.004

## Zebrafish *pcnt* mRNA is localized to the centrosome of mitotic retinal neuroepithelial cells in vivo

We next tested whether centrosomal localization of *pcnt* mRNA also takes place in differentiated tissues in vivo. We focused on the retinal neuroepithelia of 1-day-old zebrafish because at this developmental stage, retinal neuroepithelial cells in different cell cycle stages can be readily identified based on the known patterns of interkinetic nuclear migration (e.g. mitotic cells at the apical side of retina) (*Baye and Link, 2007*). Again, we observed that zebrafish *pcnt* mRNA was enriched at the centrosome of mitotic, but not of non-mitotic, neuroepithelial cells (*Figure 2—figure supplement 2*). We thus conclude that centrosomal enrichment of *pericentrin* mRNA is likely a conserved process in mitotic cells.

## Centrosomally localized *PCNT* mRNA undergoes active translation

Interestingly, the timing of this unique centrosomal accumulation of *PCNT* mRNA in cultured cells (*Figure 2*) overlaps precisely with that of centrosome maturation (*Khodjakov and Rieder, 1999*; *Piehl et al., 2004*). These observations raise the intriguing possibility that *PCNT* mRNA might be

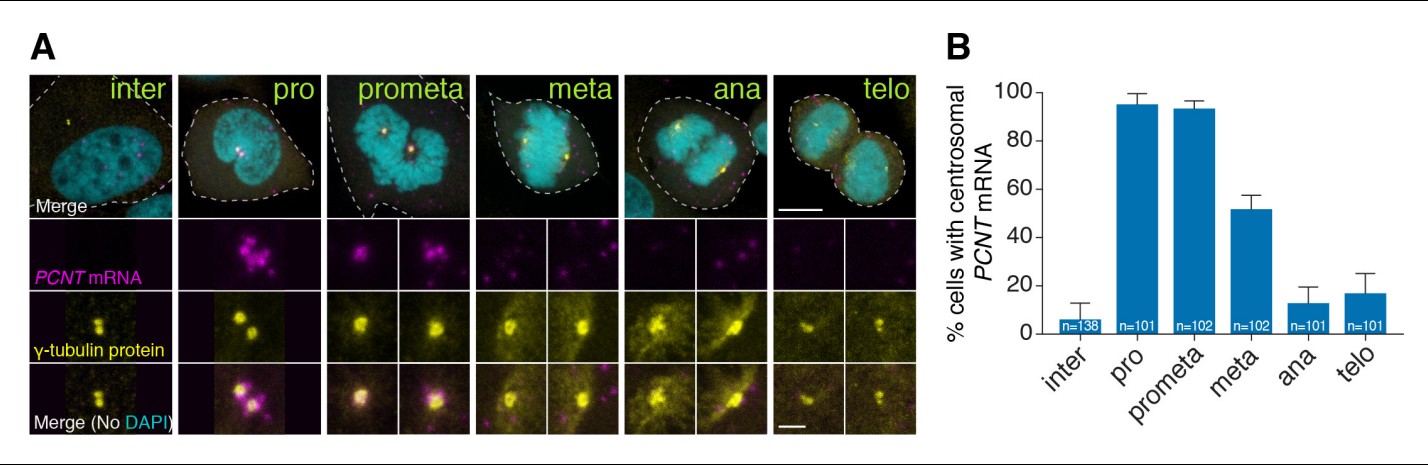

**Figure 2.** Human *PCNT* mRNA is localized to centrosomes during early mitosis. (**A**) Synchronized HeLa cells were subjected to fluorescent in situ hybridization with tyramide signal amplification against *PCNT* mRNA and anti-γ-tubulin immunostaining. Note that *PCNT* mRNA was localized to centrosomes predominantly during prophase (pro) and prometaphase (prometa). (**B**) Quantification of *PCNT* mRNA localization at centrosomes during different cell cycle stages. Data are represented as mean with standard deviation (SD) from three biological replicates, with the total number of cells analyzed indicated. Dashed lines delineate the cell boundaries. Scale bars: 10 μm and 2 μm (inset).

DOI: https://doi.org/10.7554/eLife.34959.005

The following source data and figure supplements are available for figure 2:

**Source data 1.** The source data to plot the bar chart in *Figure 2B*.

DOI: https://doi.org/10.7554/eLife.34959.008

**Figure supplement 1.** Non-overlapping antisense probes and two independent in situ methods confirm centrosomal localization of *PCNT* mRNA during early mitosis.

DOI: https://doi.org/10.7554/eLife.34959.006

**Figure supplement 2.** Zebrafish *pcnt* mRNA is localized to centrosomes of mitotic retinal neuroepithelial cells in vivo.

DOI: https://doi.org/10.7554/eLife.34959.007

translated near the centrosome to facilitate the incorporation of PCNT proteins into the PCM during centrosome maturation.

To determine whether *PCNT* mRNA is actively translated near the centrosome, we developed a strategy to detect actively translating *PCNT* polysomes by combining *PCNT* smFISH and double immunofluorescence to label *PCNT* mRNA, and the N- and C-termini of PCNT protein simultaneously (*Figure 3A*). Given the inter-ribosome distance of approximately 260 nucleotides on a transcript during translation (*Wang et al., 2016*) and the large size of *PCNT* mRNA (10 knt), a single *PCNT* transcript can be actively translated by as many as 40 ribosomes simultaneously. Therefore, up to 40 nascent polypeptides emerging from a single *PCNT* polysome can be visualized by anti-PCNT N-terminus immunostaining. By combining this immunostaining strategy with *PCNT* smFISH, multiple nascent PCNT polypeptides can be visualized on a single *PCNT* mRNA. Furthermore, the signals from antibody staining are determined by the location of the epitopes. Therefore, the translating nascent PCNT polypeptides, with the C-terminus not yet synthesized, would only show positive signals from anti-PCNT N-terminus immunostaining (and be positive for *PCNT* smFISH), whereas fully synthesized PCNT protein would show signals from both anti-PCNT N- and C-terminus immunostaining (and be negative for *PCNT* smFISH because of release of the full-length protein from the RNA-bound polysomes).

Using this strategy, we detected nascent PCNT polypeptides emerging from *PCNT* mRNA near the centrosome during early mitosis (*Figure 3B*, top panel, PCNT N$^+$/C$^-$/*PCNT* smFISH$^+$). As an important control, we showed that colocalization of *PCNT* mRNA with anti-PCNT N-terminus signals was lost after a brief treatment of cells with puromycin (*Figure 3B*, bottom panel), under a condition confirmed to inhibit translation by dissociating the ribosomes and releasing the nascent polypeptides (*Figure 3—figure supplement 1*, *Wang et al., 2016*; *Yan et al., 2016*). Next, we developed a methodology to quantify the effect of puromycin treatment on the colocalization of *PCNT* mRNA and anti-PCNT N-terminus signals in three dimensional (3D) voxels rendered from confocal z-stacks. Given that the mean radius of a mitotic centrosome is ~1 μm (*Figure 3—figure supplement 2*), we specifically quantified the fraction of *PCNT* mRNA between 1 and 3 μm from the center of each centrosome—that is, the RNA close to, but not within, the centrosome—with anti-PCNT N-terminus signals in early mitotic cells, with or without the brief puromycin treatment. Consistent with the results shown in *Figure 3B*, upon the short puromycin treatment, the fraction of *PCNT* mRNA with anti-PCNT N-terminus signals was significantly reduced, with many *PCNT* mRNA no longer bearing anti-PCNT N-terminus signals (*Figure 3C*). Furthermore, we observed that *PCNT* mRNA molecules near the centrosome were often positive for both anti-PCNT N-terminus and anti-ribosomal protein S6 signals in both HeLa and RPE-1 cells during early mitosis (*Figure 3—figure supplement 3*). Together, these results indicate that during early mitosis, a population of *PCNT* mRNA is undergoing active translation near the centrosome.

## Centrosomal localization of *pcnt/PCNT* mRNA requires intact polysomes, microtubules, and dynein activity

In addition to the loss of anti-PCNT N-terminus signals from *PCNT* mRNA, surprisingly, the brief puromycin treatment led to the population of *PCNT* mRNA shifting away from the centrosome (*Figure 4A*). Similarly, when zebrafish embryos were injected with puromycin at the one-cell stage, *pcnt* transcripts became diffused throughout the cell (*Figure 4—figure supplement 1*). Because puromycin dissociates ribosomes and nascent polypeptides, these observations suggest that *PCNT/pcnt* mRNAs in human and zebrafish are enriched near the centrosome by tethering to the actively translating ribosomes.

To further test the dependency of centrosomal enrichment of *PCNT* mRNA on intact, actively translating polysomes, we treated the cultured cells with either emetine, which stabilizes polysomes by irreversibly binding the ribosomal 40S subunit and thus 'freezing' translation during elongation (*Jiménez et al., 1977*), or harringtonine, which disrupts polysomes by blocking the initiation step of translation while allowing downstream ribosomes to run off from the mRNA (*Huang, 1975*). We found that *PCNT* mRNA localization patterns in emetine- and harringtonine-treated cells resembled those observed in vehicle- (control) and puromycin-treated cells, respectively (*Figure 4A*). Congruent with the detection of nascent PCNT polypeptides near the centrosome (*Figure 3*), these data support the model that centrosomal enrichment of *PCNT* mRNA relies on centrosomal enrichment of polysomes that are translating *PCNT* mRNA.

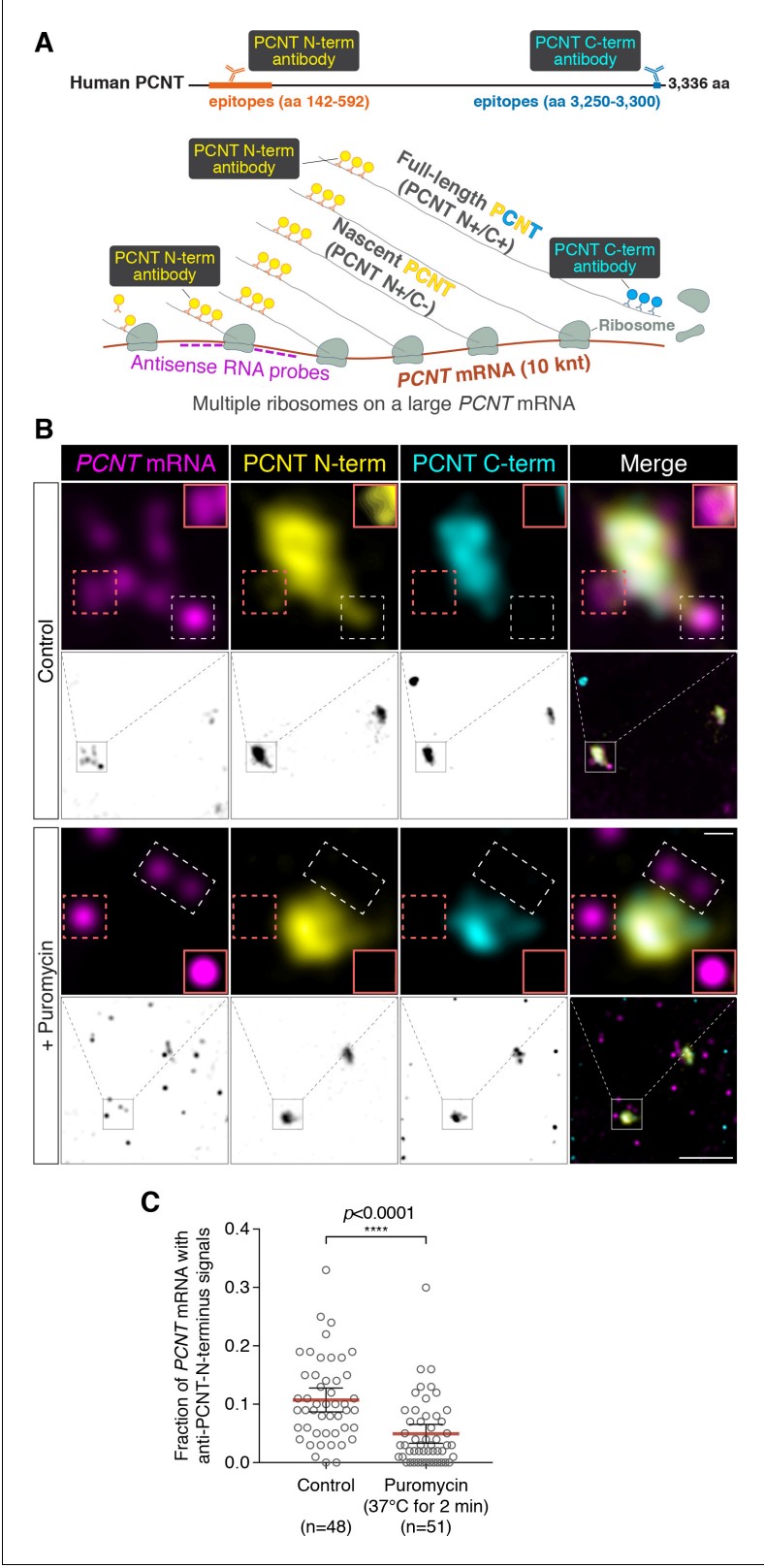

**Figure 3.** Centrosomally localized *PCNT* mRNA undergoes active translation. (**A**) A strategy of using smFISH and double immunofluorescence (IF) to distinguish between newly synthesized and full-length PCNT proteins (see text for details). The location and size of the epitopes for anti-PCNT N- and C-terminus antibodies, proportionally scaled to the full-length human PCNT protein, are indicated. (**B**) Prometaphase HeLa cells were subjected to
*Figure 3 continued on next page*

*Figure 3 continued*

*PCNT* smFISH and anti-PCNT immunostaining against the N- and C-terminus of PCNT protein (PCNT N-term and PCNT C-term). Note that the putative active translation sites were labeled by PCNT N-term IF and *PCNT* smFISH, but not by PCNT C-term IF (top panel). However, upon the puromycin treatment (300 µM for 2 min at 37°C, bottom panel), PCNT N-term IF signals were no longer colocalized with *PCNT* smFISH signals, indicating that those PCNT N-term IF signals on RNA represent nascent PCNT polypeptides. Orange boxes show higher contrast of selected areas (dashed orange boxes) for better visualization. The low-magnification images corresponding to the magnified insets are shown in monochrome (individual channels) and color (merged channels). (C) *PCNT* smFISH signals between 1 and 3 µm radius from the centrosome center were quantified for the presence of anti-PCNT N-term IF signals with or without a short puromycin treatment. Data are represented as mean ±95% CI (confidence intervals) from three biological replicates, with the total number of cells analyzed indicated. p-value was obtained with Student's t-test (two-tailed). Scale bars: 5 µm and 0.5 µm (inset).

DOI: https://doi.org/10.7554/eLife.34959.009

The following source data and figure supplements are available for figure 3:

**Source data 1.** The source data to plot the dot plot in *Figure 3C*.
DOI: https://doi.org/10.7554/eLife.34959.014
**Figure supplement 1.** Visualization of active translation in live cells using the SunTag/PP7 system.
DOI: https://doi.org/10.7554/eLife.34959.010
**Figure supplement 2.** Mean radius of mitotic centrosomes of HeLa cells.
DOI: https://doi.org/10.7554/eLife.34959.011
**Figure supplement 2—source data 1.** The source data to plot the dot plot in *Figure 3—figure supplement 2*.
DOI: https://doi.org/10.7554/eLife.34959.012
**Figure supplement 3.** Colocalization of anti-PCNT N-terminus, anti-ribosomal protein S6, and *PCNT* smFISH signals near the centrosome during early mitosis.
DOI: https://doi.org/10.7554/eLife.34959.013

We often observed that the two centrosomes in early mitotic cells were asymmetric in size where more *PCNT* mRNA was enriched near the larger centrosome (*Figure 4—figure supplement 2*). Because the microtubule nucleation activity is often positively correlated with the centrosome size, we speculated that centrosomal enrichment of *pericentrin* mRNA/polysomes might be a microtubule-dependent process. We thus tested if the localization of *pericentrin* mRNA would be perturbed when microtubules were depolymerized. We found that in both zebrafish and cultured human cells, *pcnt*/*PCNT* mRNA was no longer enriched around the centrosome upon microtubule depolymerization (*Figure 4B and C*, *Figure 4—figure supplement 3*). In contrast, a cytochalasin B treatment, which disrupts the actin cytoskeleton, had no effect on the centrosomal enrichment of *PCNT* mRNA (*Figure 4—figure supplement 3A*). These results suggest that microtubules, but not actin filaments, serve as 'tracks' on which *pericentrin* mRNA/polysomes are transported.

Given that cytoplasmic dynein is a common minus-end-directed, microtubule-based motor that transports cargo toward the microtubule minus end (i.e. toward the centrosome), we next tested whether centrosomal localization of *PCNT* mRNA is a dynein-dependent process. We treated the cells with ciliobrevin D, a specific small molecule inhibitor of cytoplasmic dynein (*Firestone et al., 2012*) and quantified the effect of this treatment on the centrosomal localization of *PCNT* mRNA. We found that *PCNT* mRNA was no longer enriched at the centrosome upon the ciliobrevin D treatment (*Figure 4D*). Together, these results indicate that centrosomal enrichment of *pericentrin* mRNA during early mitosis is a translation-, microtubule- and dynein-dependent process.

## Active translation of *PCNT* mRNA during early mitosis contributes to the optimal incorporation of PCNT protein into the mitotic PCM

To determine the functional significance of translation of centrosomally localized *PCNT* mRNA during early mitosis, we compared centrosomal PCNT levels shortly before and after mitotic entry (i.e. late G2 vs. early M phase). We arrested cultured human cells from progression out of late G2 phase using the CDK1 inhibitor RO-3306 (*Vassilev et al., 2006*). CDK1 is largely inactive during G2 and becomes activated at the onset of mitosis (*Gavet and Pines, 2010*; *Jackman et al., 2003*). In the presence of RO-3306, cells can be held at late G2 phase, and upon inhibitor washout, cells can be released into mitosis. Because cell cycle synchronization is rarely 100% homogeneous in a cell

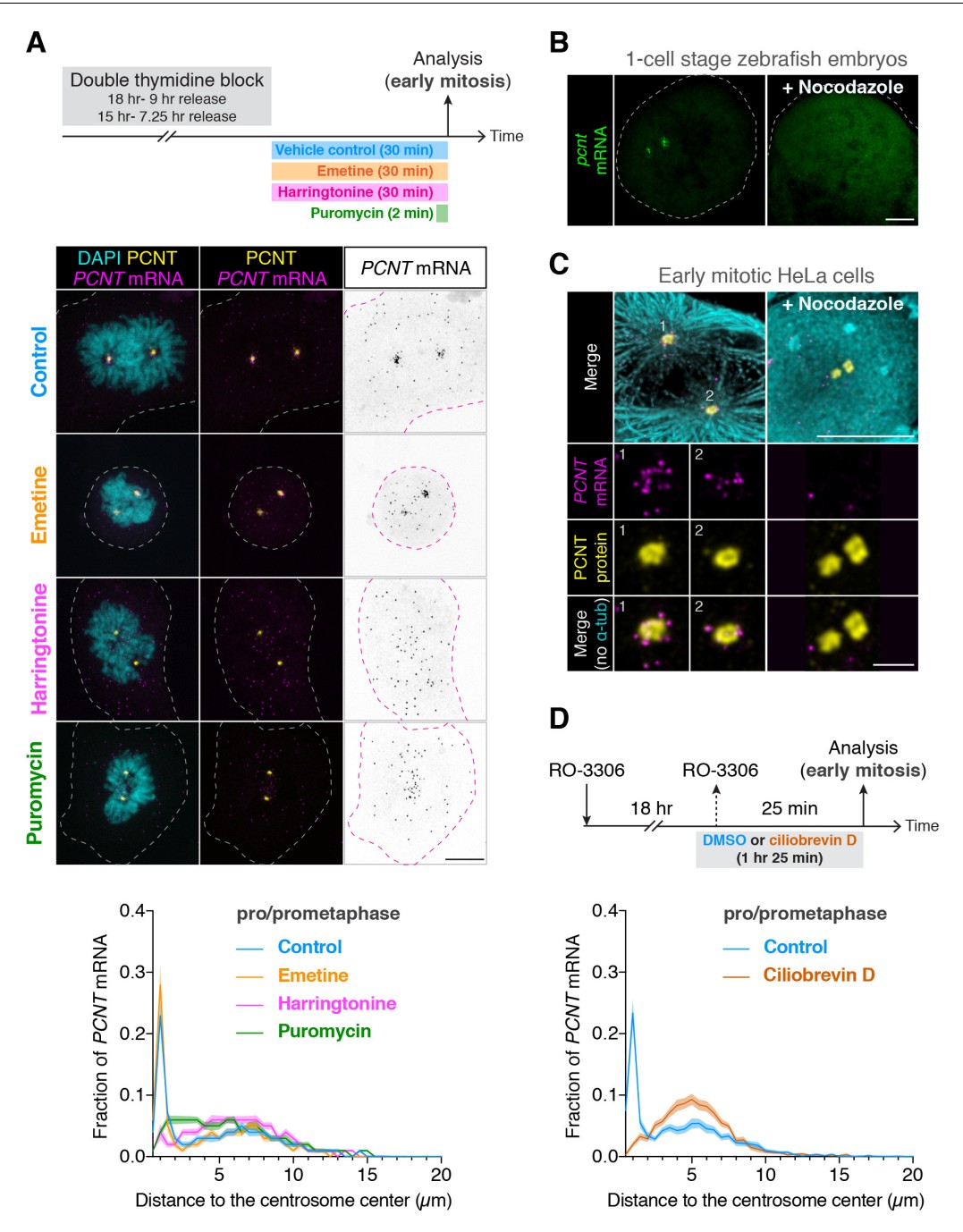

**Figure 4.** Centrosomal localization of *pcnt*/*PCNT* mRNA requires intact polysomes, microtubules, and dynein activity. (**A**) HeLa cells were synchronized by a double thymidine block and treated with DMSO vehicle (Control), 208 μM emetine, 3.76 μM harringtonine for 30 min, or 300 μM puromycin for 2 min before anti-PCNT immunostaining and *PCNT* smFISH. Representative confocal images and quantification of the *PCNT* mRNA distribution are shown for each condition. The distribution of *PCNT* mRNA in cells was quantified by measuring the distance between 3D rendered *PCNT* smFISH signals and the center of the nearest centrosome (labeled by anti-PCNT immunostaining). The fractions of mRNA as a function of distance to the nearest centrosome (binned in 0.5 μm intervals) were then plotted as mean (solid lines) ±95% CI (shading) from three biological replicates. n = 48, 45, 57, and 51 cells for control, emetine, harringtonine, and puromycin conditions, respectively. Note that *PCNT* mRNA moved away from the centrosome upon the harringtonine or puromycin treatment, but stayed close to the centrosome upon the emetine treatment, similar to the control. (**B**) Zebrafish embryos were injected with DMSO vehicle or 100 μg/ml nocodazole at the one-cell stage followed by *pcnt* FISH. (**C**) HeLa cells were treated with DMSO vehicle or 3 μg/ml nocodazole for 2 hr at 37°C before anti-α-tubulin, anti-PCNT immunostaining, and *PCNT* smFISH. Note that *pcnt*/*PCNT* mRNA in early embryos (**B**) and in early mitotic cells (**C**) was no longer enriched at the centrosome after microtubules were depolymerized. (**D**) HeLa cells were synchronized by RO-3306 and treated with DMSO vehicle or 50 μM ciliobrevin D for 1 hr 25 min before anti-PCNT immunostaining and *PCNT*

*Figure 4 continued on next page*

*Figure 4 continued*

smFISH. The distribution of *PCNT* mRNA in cells was quantified as in (**A**). n = 63 and 70 cells for control and ciliobrevin D conditions, respectively, from a representative experiment (two technical duplicates per condition). Note that *PCNT* mRNA was no longer enriched at the centrosome upon the ciliobrevin D treatment. Dashed lines delineate the cell boundaries. Scale bars, 10 μm (**A**), 100 μm (**C**), 10 μm (**D**), and 2 μm (inset in **D**).

DOI: https://doi.org/10.7554/eLife.34959.015

The following source data and figure supplements are available for figure 4:

**Source data 1.** The source data to plot the dot plots in *Figure 4A and 4D*.
DOI: https://doi.org/10.7554/eLife.34959.021
**Figure supplement 1.** Centrosomal localization of zebrafish *pcnt* mRNA depends on intact polysomes.
DOI: https://doi.org/10.7554/eLife.34959.016
**Figure supplement 2.** More *PCNT* mRNA was often enriched near the larger centrosome in early mitosis.
DOI: https://doi.org/10.7554/eLife.34959.017
**Figure supplement 2—source data 1.** The source data to plot *Figure 4—figure supplement 2*.
DOI: https://doi.org/10.7554/eLife.34959.018
**Figure supplement 3.** Centrosomal localization of human *PCNT* mRNA during early mitosis is microtubule-dependent.
DOI: https://doi.org/10.7554/eLife.34959.019
**Figure supplement 3—source data 1.** The source data to plot the *PCNT* mRNA distribution, PCNT protein distribution, and PCNT protein intensities in *Figure 4—figure supplement 3B*
DOI: https://doi.org/10.7554/eLife.34959.020

population, we decided to quantify the amount of centrosomal PCNT at the single-cell level using anti-PCNT immunostaining of individual cells. To confidently identify late G2 cells in RO-3306-treated population, we used a RPE-1 cell line stably expressing Centrin-GFP (*Uetake et al., 2007*) and categorized the cells as 'late G2' if (1) their two centrosomes (with two centrin dots per centrosome) were separated by >2 μm—a sign indicating the loss of centrosome cohesion that occurs during late G2 to M transition (*Bahe et al., 2005*; *Fry et al., 1998*; *Mardin et al., 2011*) and (2) their DNA was not condensed. We identified the cells as early M phase cells (i.e. prophase or prometaphase) 25 min after RO-3306 washout by observing DNA morphology.

Using this strategy, we found that approximately twofold more PCNT proteins were incorporated into the centrosomes in early mitotic cells as compared to late G2 cells (*Figure 5A*). Importantly, the numbers of *PCNT* mRNA did not significantly differ between late G2 and early M phases, even though there was an approximately fourfold increase from G1 to late G2 phases (*Figure 5B*). Therefore, these results indicate that the increase in centrosomal PCNT protein levels when cells progress from G2 to M phases (e.g., the 25-min period after RO-3306 washout) is due to upregulation of translation and not to altered mRNA abundance.

To independently assess the impact of translation during early mitosis on PCNT incorporation into the centrosomes, we disrupted this process by pulsing the RO-3306 synchronized cells with puromycin to inhibit translation for 2 min, followed by immediate fixation and anti-PCNT immunostaining. As previously shown, this condition inhibits translation acutely and dissociates PCNT nascent polypeptides from *PCNT* mRNA-containing polysomes, including those near the centrosome (*Figure 3*). We found that in the puromycin-treated cells,~30% fewer PCNT molecules were incorporated into the PCM than in the control cells during prophase/prometaphase (*Figure 5C*). These results indicate that active translation during prophase/prometaphase is required for efficient incorporation of PCNT into the mitotic centrosomes; disruption of this process, even just briefly, significantly affects the PCNT level at the centrosomes.

Collectively, these results indicate that active translation of *PCNT* mRNA during early mitosis contributes to the optimal incorporation of PCNT proteins into the mitotic PCM and that this is most plausibly achieved by co-translational targeting of the *PCNT* mRNA-containing polysomes to the proximity of the mitotic centrosomes.

## *ASPM* mRNA is enriched at the centrosome in a translation-dependent manner during mitosis

To determine if the cell uses a similar co-translational targeting strategy to target other large proteins to the centrosome, we examined the distribution of *CEP192*, *CEP215/CDK5RAP2*, and *ASPM* mRNA in cultured human cells. We found that while *CEP192* and *CEP215* mRNA did not show any

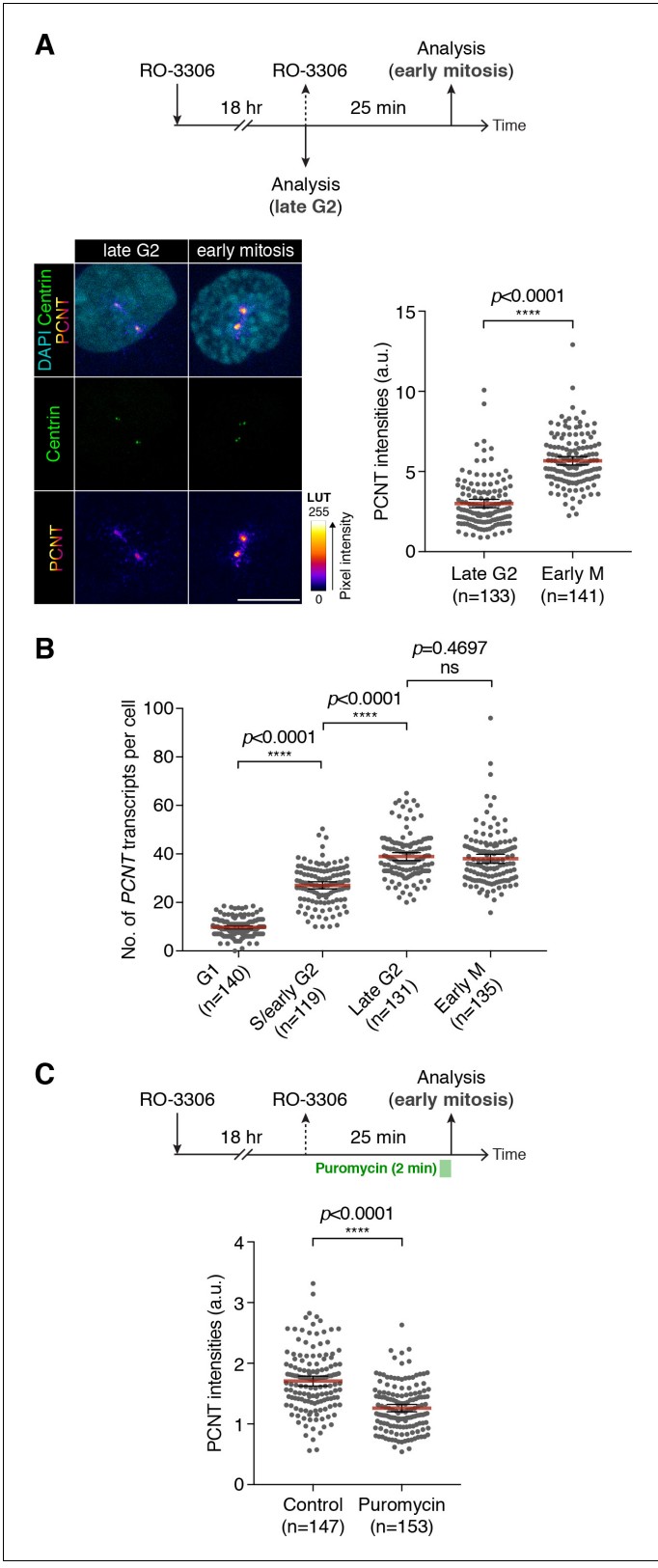

**Figure 5.** Centrosomal localization of *PCNT* mRNA/polysomes contributes to PCNT incorporation into mitotic centrosomes. (**A**) Centrin-GFP RPE-1 cells—at either late G2 or early M phase—were subjected to anti-PCNT immunostaining. Representative confocal images are shown for each condition. A 'fire' lookup table (LUT) was used to show PCNT signal intensities. The sum intensity of anti-PCNT signals from both centrosomes of each cell *Figure 5 continued on next page*

*Figure 5 continued*

was measured and plotted. (**B**) Numbers of *PCNT* mRNA at different cell cycle stages of Centrin-GFP RPE-1 cells were determined by *PCNT* smFISH. S phase/early G2 cells were identified by EdU labeling for 30 min. (**C**) HeLa cells were treated with vehicle control or 300 μM puromycin for 2 min before anti-PCNT immunostaining. The sum intensity of anti-PCNT signals from both centrosomes of each prophase or prometaphase cell was measured and plotted. Data are represented as mean ±95% CI. 'n' indicates the total number of cells analyzed from two (**A**), three (**B**), and two (**C**) biological replicates. p-values were obtained with Student's t-test (two-tailed). a.u., arbitrary unit. Scale bar: 10 μm.

DOI: https://doi.org/10.7554/eLife.34959.022

The following source data is available for figure 5:

**Source data 1.** The source data to plot the dot plots in *Figure 5A–C*.

DOI: https://doi.org/10.7554/eLife.34959.023

centrosomal enrichment during early mitosis (data not shown), *ASPM* mRNA was strongly enriched at the centrosome during prometaphase and metaphase in both RPE-1 and HeLa cells (*Figure 6* and *Figure 6—figure supplement 1*). Furthermore, upon a short puromycin treatment, *ASPM* mRNA became dispersed throughout the cell, indicating that centrosomal enrichment of *ASPM* mRNA also requires intact polysomes as in the case with *PCNT* mRNA. ASPM (and its fly ortholog Asp) is not a PCM component per se, but a microtubule minus-end regulator (*Jiang et al., 2017*) and a spindle-pole focusing factor (*Ito and Goshima, 2015*; *Ripoll et al., 1985*; *Tungadi et al., 2017*). It is highly enriched at the mitotic spindle poles, particularly from early prometaphase to metaphase (*Ito and Goshima, 2015*; *Jiang et al., 2017*; *Tungadi et al., 2017*). Therefore, these data demonstrate

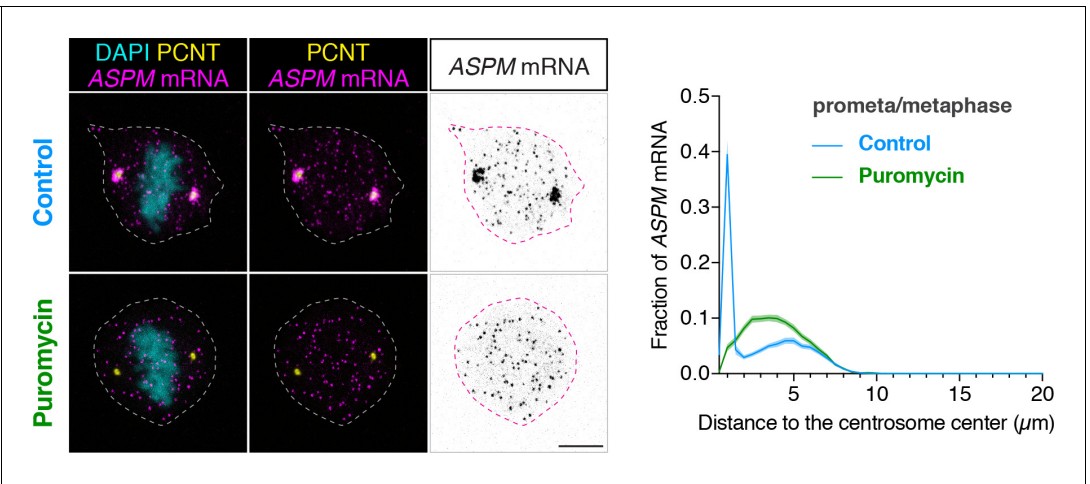

**Figure 6.** *ASPM* mRNA is enriched at centrosomes in a translation-dependent manner during mitosis. Prometaphase/metaphase RPE-1 cells were treated with vehicle (Control) or 300 μM puromycin for 2 min at 37°C (Puromycin) before fixation, followed by anti-PCNT immunostaining and *ASPM* smFISH. Representative confocal images and quantification of the *ASPM* mRNA distribution are shown for each condition. The distribution of *ASPM* mRNA in cells was quantified by measuring the distance between 3D rendered *ASPM* smFISH signals and the center of the nearest centrosome (labeled by anti-PCNT immunostaining). The fractions of mRNA as a function of distance to the nearest centrosome (binned in 0.5 μm intervals) were then plotted as mean (solid lines)±95% CI (shading) from two biological replicates. n = 76 and 81 cells for control and puromycin conditions, respectively. Note that *ASPM* mRNA was enriched at the centrosomes/spindle poles of the metaphase cell, but became dispersed throughout the cell upon a short puromycin treatment. Scale bars: 10 μm.

DOI: https://doi.org/10.7554/eLife.34959.024

The following source data and figure supplements are available for figure 6:

**Source data 1.** The source data to plot the histogram in *Figure 6*.

DOI: https://doi.org/10.7554/eLife.34959.027

**Figure supplement 1.** *ASPM* mRNA is enriched at centrosomes in a translation-dependent manner during mitosis.

DOI: https://doi.org/10.7554/eLife.34959.025

**Figure supplement 1—source data 1.** The source data to plot the histogram in *Figure 6—figure supplement 1*.

DOI: https://doi.org/10.7554/eLife.34959.026

another example of spatiotemporal coupling between active translation and translocation of polysomes to the final destination of the protein being synthesized.

## Discussion

Here, we report that PCNT protein is delivered co-translationally to the centrosome during centrosome maturation through a microtubule- and dynein-dependent process. This process is demonstrated by centrosomal enrichment of *PCNT* mRNA, its translation near the centrosome, and requirement of intact translation machinery for *PCNT* mRNA localization during early mitosis. The translation- and microtubule-dependent centrosomal enrichment of *pericentrin* mRNA is observed in both zebrafish embryos and human somatic cell lines. Interestingly, the mRNA of the sole *pcnt* ortholog, *plp*, of *Drosophila melanogaster*, was also previously reported to localize to the centrosome in early fly embryos (*Lécuyer et al., 2007*). Although it has not been shown if the centrosomally localized *plp* mRNA undergoes active translation, it is tempting to speculate that co-translational targeting of PCNT (and its orthologous proteins) to the centrosome is an evolutionarily conserved process. In addition to PCNT, the cell appears to use a similar co-translational targeting strategy to deliver the large microtubule minus-end regulator/spindle-pole focusing factor, ASPM, to mitotic spindle poles, as *ASPM* mRNA is strongly enriched at mitotic spindle poles in a translation-dependent manner, concomitantly with the ASPM protein level reaching its maximum at the same place. We suspect that co-translational targeting of polysomes translating a subset of cytoplasmic proteins to specific subcellular destinations is a widespread mechanism used in post-transcriptional gene regulation.

### Evidence supporting translation of *PCNT* mRNA near the centrosome

In this study, we also developed a strategy of visualizing active translation. We took advantage of the large size of *PCNT* mRNA and combined *PCNT* smFISH and immunofluorescence against the N- or C-terminal epitopes of PCNT nascent polypeptides to detect which *PCNT* mRNA molecules were undergoing active translation (*Figure 3*). This imaging-based method allowed us to determine whether the PCNT was being newly synthesized 'on site' or the PCNT was made somewhere within the cell and then transported/diffused to the centrosome because only the former would show positive signals for N-, but not C-terminus immunostaining of the synthesized protein, and these signals would be sensitive to the puromycin treatment. However, detecting nascent PCNT polypeptides by anti-PCNT N-terminus antibody staining relies on multiple copies of polypeptides tethered to the translating ribosomes for generating detectable fluorescent signals. Therefore, this method is biased toward detecting the translating *PCNT* polysomes at later stages of translation elongation, when multiple ribosomes have been loaded and multiple copies of PCNT polypeptides are available for antibody detection. This method, however, would likely fail to detect anti-PCNT N-terminus signals on the mRNA that just started to be translated. We speculate that this could explain why not all centrosomally localized *PCNT* mRNAs showed anti-PCNT N-terminus signals, although most of these *PCNT* mRNAs would shift away from the centrosome upon the puromycin or harringtonine treatment (*Figure 4*). Translation of *PCNT* mRNA near the centrosome is further supported by the co-localization of ant-PCNT N-terminus, anti-ribosomal protein S6, and *PCNT* smFISH signals near the centrosome during early mitosis in two different human cell lines (*Figure 3—figure supplement 3*). Together, these multiple lines of evidence—(1) co-localization of anti-PCNT N-terminus but not anti-PCNT C-terminus signals with *PCNT* mRNA, (2) puromycin-sensitive anti-PCNT N-terminus/*PCNT* mRNA signals, (3) polysome-dependent centrosomal enrichment of *PCNT* mRNA, and (4) co-localization of PCNT N-terminus/*PCNT* mRNA signals with a ribosomal protein—strongly support the conclusion that *PCNT* mRNA is locally translated near the centrosome during early mitosis.

### Mechanisms of co-translational targeting and centrosome maturation

How are the polysomes actively translating PCNT or ASPM targeted to the centrosome? In the case of PCNT, previous studies have shown that PCNT protein is transported to the centrosome through its interaction with cytoplasmic dynein (*Purohit et al., 1999*; *Young et al., 2000*), specifically through the dynein light intermediate chain 1 (LIC1) (*Tynan et al., 2000*). Moreover, the LIC1-interacting domain in PCNT is mapped within ~550 amino acids located in the N-terminal half of PCNT (*Tynan et al., 2000*). Based on these findings, we propose a model in which the partially translated

PCNT nascent polypeptide starts to interact with the dynein motor complex once the LIC1-interacting domain in the N-terminal half of PCNT is synthesized and folded, as early stages of protein folding can proceed quickly and co-translationally (*Fedorov and Baldwin, 1997*; *Komar et al., 1997*; *Ptitsyn, 1995*; *Roder and Colón, 1997*). Subsequently, this nascent polypeptide-dynein interaction allows the entire polysome, which is still actively translating *PCNT* mRNA, to be transported along the microtubule toward the centrosome (*Figure 7*). Alternatively, it is also possible that the coupling of the polysome to the motor complex is mediated through the ribosome-dynein interaction. If this was the case, additional components/adaptors would need to be involved in the interaction to differentiate the ribosomes translating *PCNT* mRNA from the ones translating other transcripts. One of the above mechanisms (i.e. via interaction through the nascent chain or ribosome itself) may also be used to mediate the co-translational targeting of *ASPM* mRNA/polysomes to the mitotic spindle poles. Mapping the binding domains on both the motor and cargo sides, identifying the cargo adapter(s) that mediates the interaction, and testing the roles of mitotic kinases that are known to be involved in centrosome maturation such as Aurora A and PLK1 (*Glover et al., 1998*; *Hannak et al., 2001*; *Petronczki et al., 2008*) are important next steps to dissect the mechanisms underlying this co-translational protein targeting process. Among the mitotic kinases that could be directly involved in this process, PLK1 is an attractive candidate for the following reasons:

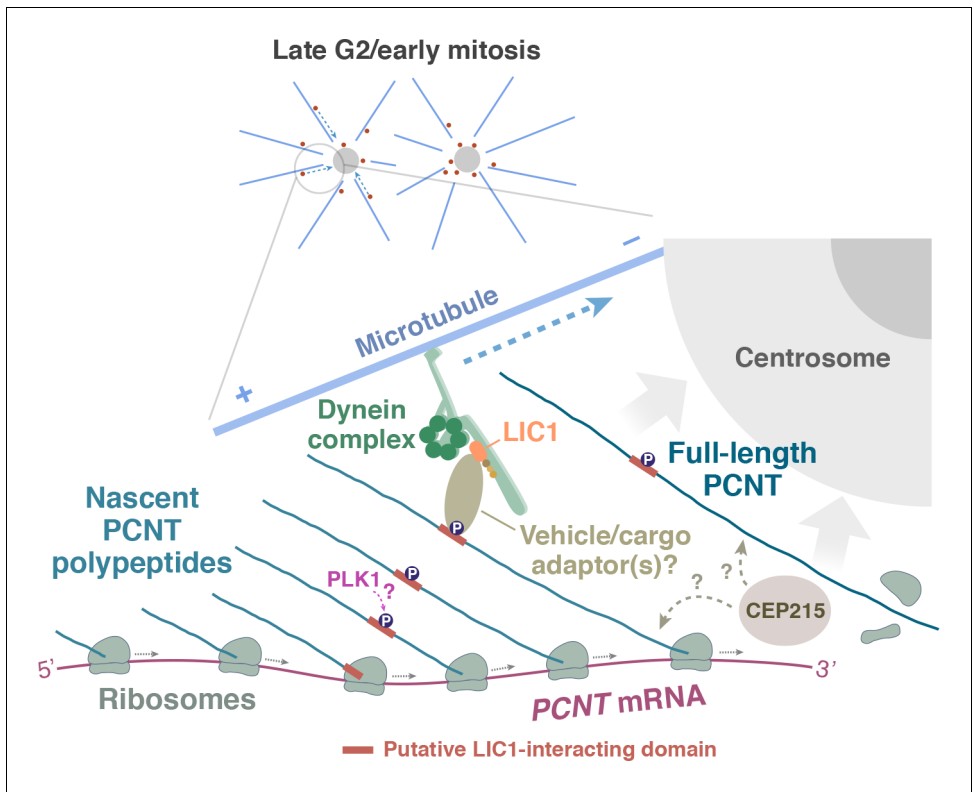

**Figure 7.** A model of co-translational targeting of *PCNT* polysomes toward the centrosome during centrosome maturation. During the late G2/M transition, translation of *PCNT* mRNA is upregulated by an as yet unknown mechanism. The partially translated PCNT nascent polypeptide starts to interact with the dynein motor complex once the dynein light intermediate chain 1 (LIC1)-interacting domain in the N-terminal half of PCNT is synthesized and folded. It will be interesting to test if PLK1 phosphorylation of S1235 and S1241 within the LIC1-interacting domain initiates this PCNT-dynein interaction. Subsequently, this nascent polypeptide-dynein interaction allows the entire polysome, which is still actively translating *PCNT* mRNA, to be transported along the microtubule toward the centrosome. This co-translational targeting mechanism may maximize efficiency of PCNT production and delivery to the centrosome, prevent ectopic accumulation of PCNT outside of centrosomes, and/or facilitate integration of PCNT into the expanding PCM during early mitosis. It remains to be determined if other PCM components (e.g. CEP215) interact with PCNT co- and/or post-translationally.
DOI: https://doi.org/10.7554/eLife.34959.028

Phosphorylation of human PCNT at S1235 and S1241 by PLK1 is required for the recruitment of several other PCM proteins for centrosome maturation (*Lee and Rhee, 2011*). In addition, inhibition of PLK1 activity also reduces PCNT levels at mitotic centrosomes (*Haren et al., 2009*; *Lee and Rhee, 2011*; *Santamaria et al., 2007*). Notably, these two PLK1 phosphorylation sites, S1235 and S1241, are highly conserved among the vertebrates and are located within the putative LIC1-binding domain that interacts with cytoplasmic dynein (Tynan et al., 2000). It is thus tempting to speculate that PLK1-dependent phosphorylation of these two conserved residues might be required for mediating the PCNT-dynein interaction and thus initiating co-translational targeting of PCNT to centrosomes.

Our finding that new PCNT is delivered co-translationally to the centrosome during centrosome maturation also raises an important question of when and how PCNT interacts with other PCM components that are also required for centrosome maturation such as CEP192 and CEP215 (*Barr et al., 2010*; *Choi et al., 2010*; *Gomez-Ferreria et al., 2007*; *Joukov et al., 2014*; *Kim and Rhee, 2014*; *Zhu et al., 2008*). For example, vertebrate PCNT and CEP215 interact and depend on each other for localizing to mitotic centrosomes (*Buchman et al., 2010*; *Haren et al., 2009*; *Kim and Rhee, 2014*; *Lawo et al., 2012*). However, zebrafish Cep215 and human CEP215 may not be targeted to centrosomes co-translationally because their transcripts do not show centrosomal enrichment during early mitosis (*Figure 1* and data not shown). It thus remains unclear when and where the PCNT-CEP215 interaction occurs and if this interaction takes place co- and/or post-translationally. Determining if the translating *PCNT* polysomes contain CEP215 proteins could be the first step to distinguish these possibilities. Clearly, it will be important to elucidate how co-translational targeting of PCNT (and possible other PCM components) fits in with the current model of centrosome maturation that involves the interplay of several other PCM proteins.

## Significance of co-translational targeting of PCNT to the centrosome during mitosis

What might be the biological significance of co-translational targeting of unusually large proteins such as PCNT or ASPM to the centrosome during mitosis? In the case of PCNT, we propose three nonexclusive possibilities. First, since PCNT has been placed upstream as a scaffold to initiate centrosome maturation (*Lee and Rhee, 2011*) and to help recruit other PCM components, including NEDD1, CEP192, and CEP215/CDK5RAP2 (*Lawo et al., 2012*; *Lee and Rhee, 2011*), it is critical to have optimal amounts of PCNT incorporated at the centrosome early during mitosis. Because dynein-mediated cargo transport is relatively fast, typically ranging from 0.5 to 3 µm per second in vivo (*Schlager et al., 2014*; *Yang et al., 2007*), it seems that PCNT protein molecules can be transported from anywhere in the cell to the centrosome in seconds, regardless whether they are in a polysome or not. However, dynein cargos in cells are likely powered by several dynein motors at a time (*Kardon and Vale, 2009*) and the large PCNT protein requires 10–20 min to synthesize. Therefore, mechanistically coupling translation and translocation of polysomes containing multiple copies of nascent PCNT polypeptides could help the cell not only use the dynein motor pool economically but also enhance transport efficiency. Second, generating PCNT proteins elsewhere in the cell might be deleterious. For example, non-centrosomal accumulation of PCNT might recruit other PCM components to the unwanted locations, resulting in ectopic PCM assembly, as PCNT overexpression induces a marked increase in centrosome size and the recruitment of other PCM proteins (*Lawo et al., 2012*; *Loncarek et al., 2008*). Co-translational targeting of PCNT on defined microtubule tracks through the dynein motor can help confine most full-length PCNT proteins to the centrosome. Consistent with this view, we observed that if microtubules were depolymerized before mitosis, not only was less PCNT incorporated into mitotic centrosomes, a portion of PCNT also became dispersed throughout the cytoplasm as small PCNT puncta (*Figure 4—figure supplement 3B*). This result implies that full-length PCNT synthesized in the cytoplasm was not incorporated into centrosomes efficiently without the microtubule-mediated, co-translational protein targeting. Third, co-translational targeting of nascent PCNT polypeptides might be an integrated part of mitotic PCM expansion. Akin to the co-translational targeting of membrane and secreted proteins to the endoplasmic reticulum (ER), where the translating nascent polypeptides undergo protein folding and post-translational modifications in the ER lumen (*Bergman and Kuehl, 1979*; *Chen et al., 1995*), co-translational targeting of nascent PCNT polypeptides might promote their proper folding and complex formation near the PCM, thereby facilitating integration into the expanding PCM during early

mitosis. Another possible mechanism by which co-translational targeting may facilitate PCNT integration into the PCM is through the process of liquid-liquid phase separation. The centrosome is a membrane-less organelle in which the PCM has liquid-like properties. Emerging evidence suggests that such an organelle may be formed by phase separation of compartments into 'biomolecular condensates' (*Banani et al., 2017*). Indeed, purified SPD-5, a key mitotic PCM component with extensive coiled-coil domains in *C. elegans*, can phase separate into spherical condensates that recruit microtubule nucleating proteins, tubulin, and form microtubule asters, mimicking the properties of in vivo PCM (*Woodruff et al., 2017*). In addition, ribonucleoprotein granules can also phase separate into dynamic liquid droplets in vitro (*Lin et al., 2015*; *Patel et al., 2015*). Given that PCNT is a large protein with numerous coiled-coil domains and is targeted to mitotic PCM as a large ribonucleoprotein complex (polysome), it will be fascinating in the future to determine whether co-translational targeting of *PCNT* polysomes to the centrosome could be part of a phase-separation process that promotes the integration of newly synthesized PCNT proteins into the expanding PCM.

## Mitotic translation regulation of PCNT

Our data also underscore the importance of active translation of *PCNT* mRNA during early mitosis for the centrosome to gain the optimal level of PCNT because (1) during the G2/M transition, *PCNT* mRNA levels remain largely constant, but the centrosomal PCNT protein levels increase ~two fold in 25 min after the onset of mitosis; (2) inhibiting translation briefly during early mitosis—for example, 2 min of puromycin treatment in prophase or prometaphase—is sufficient to substantially reduce the amount of PCNT proteins incorporated at centrosomes (*Figure 5*).

It is still unclear how the translation activation of *PCNT* mRNA is regulated during early mitosis. Previous studies show that translation is globally repressed during mitosis (*Bonneau and Sonenberg, 1987*; *Fan and Penman, 1970*; *Pyronnet et al., 2000*), and this global translation repression is accompanied by the translation activation of a subset of transcripts through a cap-independent translation initiation mediated by internal ribosome entry sites (IRESes) (*Cornelis et al., 2000*; *Marash et al., 2008*; *Pyronnet et al., 2000*; *Qin and Sarnow, 2004*; *Ramírez-Valle et al., 2010*; *Schepens et al., 2007*; *Wilker et al., 2007*). However, a recent study has challenged this view of IRES-dependent translation during mitosis and instead finds that canonical, cap-dependent translation still dominates in mitosis as in interphase (*Shuda et al., 2015*). Therefore, to elucidate the mechanism underlying the translation upregulation of *PCNT* mRNA during early mitosis, determining if this process is a cap- and/or IRES-dependent process might be a first logical step. In addition, our recent study has linked GLE1, a multifunctional regulator of DEAD-box RNA helicases, to the regulation of PCNT levels at the centrosome (*Jao et al., 2017*). Since all known functions of GLE1 are to modulate the activities of DEAD-box helicases in mRNA export and translation (*Alcázar-Román et al., 2006*; *Bolger et al., 2008*; *Bolger and Wente, 2011*; *Weirich et al., 2006*), it is worth elucidating whether translation upregulation of *PCNT* mRNA during mitosis is regulated through the role of GLE1 in modulating certain DEAD-box helicases involved in translation control such as DDX3 (*Chen et al., 2016*; *Lai et al., 2008*; *Soto-Rifo et al., 2012*).

## A new mode of protein targeting

Protein targeting to subcellular localization via mRNA localization has been widely used in many other biological contexts. For example, in *Drosophila* and *Xenopus* oocytes, segregation of cell fates and embryonic patterning are driven by asymmetrically distributed fate determinants in the form of localized mRNA (*Bashirullah et al., 1998*; *Deshler et al., 1998*; *Ephrussi et al., 1991*). In *Saccharomyces cerevisiae*, mating type switching is regulated by targeting *ASH1* mRNA to the bud tip, where Ash1 protein is translated and acts as a repressor of mating type switching (*Long et al., 1997*; *Takizawa et al., 1997*). In fibroblasts, localizing *β-actin* mRNA to the leading edge, coupled to its local translation, promotes local actin assembly and directional migration (*Hill et al., 1994*; *Sundell and Singer, 1991*). Similarly, in neurons, many mRNAs are axonally and dendritically enriched; local translation of a subset of these mRNAs allows synapse-restricted responses to environmental cues (*Lin and Holt, 2007*; *Sutton and Schuman, 2006*; *Wu et al., 2005*). However, unlike the co-translational targeting of *PCNT* and *ASPM* mRNA to the centrosome described here, in most of the above examples, the mRNAs are transported in a translation-repressed state before arriving their destinations. For the proteins targeted to ER for the secretory pathway, translation is also

arrested before the mRNA-ribosome-nascent chain complex reaches the destined membrane, where co-translational translocation of the polypeptide into the ER resumes (*Cross et al., 2009*; *Keenan et al., 2001*). A similar ER-like co-translational translocation mechanism is also used for importing a subset of mitochondrial proteins (*Verner, 1993*; *Yogev et al., 2007*). Therefore, in contrast to all the above examples, we have described a new version of co-translational protein targeting mechanism in which mRNA targeting and translation take place simultaneously. In support of this new protein targeting mechanism, a recent study using a live translation reporter shows that reporter mRNA can be actively translated while being transported in neurons (*Wu et al., 2016*). An important next step is to determine how widely this new mode of protein targeting is employed and how it contributes to a broad context of spatially restricted gene expression.

In summary, the work presented here shows that incorporating PCNT into the PCM during centrosome maturation is at least in part mediated by upregulation of PCNT translation during the G2/M transition and the co-translational targeting of translating *PCNT* polysomes toward the centrosome during early mitosis. Efforts so far on elucidating the mechanism underlying centrosome maturation has focused for the most part on the interplay of protein-protein interactions and post-translational modifications (e.g. phosphorylation) of different PCM components. However, our study suggests that a spatiotemporal coupling between the active translation machinery and the motor-based transport may represent a new layer of control over centrosome maturation. Our work also suggests that spatially restricted mRNA localization and translation are not limited to early embryos or specialized cells (e.g. polarized cells such as neurons). We anticipate that co-translational protein targeting to subcellular compartments beyond the centrosome may prove to be a recurrent cellular strategy to synthesize and deliver certain cytoplasmic proteins to the right place at the right time. This regulatory process might represent an underappreciated, universal protein targeting mechanism, in parallel to the evolutionarily conserved co-translational targeting of secreted and membrane proteins to the ER for the secretory pathway.

# Materials and methods

## Key resources table

| Reagent type | Reagent | Source | Cat. no. | Additional information |
|---|---|---|---|---|
| Chemical compound, drug | RO-3306 | R and D Systems, Minneapolis, MN | 4181 | |
| Chemical compound, drug | Ciliobrevin D | MilliporeSigma, Burlington, MA | 250401 | |
| Chemical compound, drug | Nocodazole | Sigma-Aldrich, St. Louis, MO | M1404 | |
| Chemical compound, drug | Cytochalasin B | ACROS Organics, Geel, Belgium | 228090250 | |
| Chemical compound, drug | Cycloheximide | Alfa Aesar, Tewksbury, MA | J66901 | |
| Chemical compound, drug | Emetine | MilliporeSigma, Burlington, MA | 324693 | |
| Chemical compound, drug | Puromycin | MilliporeSigma, Burlington, MA | 540222 | |
| Chemical compound, drug | Harringtonine | LKT Laboratories, St. Paul, MN | H0169 | |
| Antibody | Rabbit anti-PCNT N terminus | Abcam, Cambridge, MA | Abcam Cat# ab4448, RRID:AB_304461 | 1:500 or 1:1000 dilution |
| Antibody | Goat anti-PCNT C terminus | Santa Cruz Biotechnology Inc., Santa Cruz, CA | Santa Cruz Biotechnology Cat# sc-28145, RRID:AB_2160666 | 1:500 dilution |
| Antibody | Mouse anti-γ-tubulin | Sigma-Aldrich, St. Louis, MO | Sigma-Aldrich Cat# T6557, RRID:AB_477584 | 1:1000 dilution |

*Continued on next page*

*Continued*

| Reagent type | Reagent | Source | Cat. no. | Additional information |
|---|---|---|---|---|
| Antibody | Rabbit anti-phospho-Histone H3(Ser10) | MilliporeSigma, Burlington, MA | MilliporeSigma Cat# 06–570, RRID:AB_310177 | 1:500 dilution |
| Antibody | Mouse anti-ribosomal protein S6 | Santa Cruz Biotechnology Inc., Santa Cruz, CA | Santa Cruz Biotechnology Cat# sc-28145, RRID:AB_1129205 | 1:500 dilution |
| Antibody | Sheep anti-digoxigenin -alkaline phosphatase | Roche Diagnostics, Mannheim, Germany | Roche Cat# 11093274910, RRID:AB_514497 | 1:5000 dilution |
| Antibody | Sheep anti-digoxigenin -peroxidase | Roche Diagnostics, Mannheim, Germany | Roche Cat# 11207733910, RRID:AB_514500 | 1:500 dilution |
| Commercial assay or kit | MEGAshortscript T7 kit | Thermo Fisher Scientific, Waltham, MA | AM1354 | |
| Commercial assay or kit | mMESSAGE mMACHINE T3 kit | Thermo Fisher Scientific, Waltham, MA | AM1348 | |
| Commercial assay or kit | Click-iT EdU Imaging Kit | Life Technologies, Carlsbad, CA | C10337 | |
| Model organism | Wild-type NHGRI-1 fish | A gift from Shawn Burgess, NHGRI/NIH, Bethesda, MA | ZIRC Cat# ZL12751, RRID:ZIRC_ZL12751 | |
| Model organism | pcnt$^{tup2}$ fish | This study | | |
| Model organism | pcnt$^{tup5}$ fish | This study | | |
| Cell line | HeLa cells | ATCC CCL-2. A gift from Susan Wente, Vanderbilt University, Nashville, TN | ATCC Cat# CCL-2, RRID:CVCL_0030 | |
| Cell line | RPE-1 cells | A gift from Irina Kaverina, Vanderbilt University, Nashville, TN | ATCC Cat# CRL-4000, RRID:CVCL_4388 | |
| Cell line | HeLa cells stably expressing scFv-sfGFP-GB1 and NLS-tdPCP-tdTomato | A gift from Xiaowei Zhuang, Howard Hughes Medical Institute, Harvard University, Cambridge, MA | | |
| Cell line | RPE-1 cells expressing Centrin-GFP | A gift from Alexey Khodjakov, Wadsworth Center, 485 New York State Department of Health, Rensselaer Polytechnic Institute, Albany, NY | | |
| Software | Huygens Professional | Scientific Volume Imaging b.v., Hilversum, Netherlands | Huygens Software, RRID:SCR_014237 | |
| Software | Imaris | Bitplane, Belfast, UK | Imaris, RRID:SCR_007370 | |
| Software | MATLAB | MathWorks, Natick, MA | MATLAB, RRID:SCR_001622 | |
| Software | Prism 7 | GraphPad, CA | Graphpad Prism, RRID:SCR_002798 | |

## Zebrafish husbandry

Wild-type NHGRI-1 fish (*LaFave et al., 2014*) were bred and maintained using standard procedures (*Westerfield, 2000*). Embryos were obtained by natural spawning and staged as described (*Kimmel et al., 1995*). All animal researches were approved by the Institutional Animal Care and Use Committee, Office of Animal Welfare Assurance, University of California, Davis.

## Generation of *pcnt* knockout fish

Disruption of zebrafish *pcnt* was done by the CRISPR-Cas technology as described (*Jao et al., 2013*). In brief, to generate guide RNA (gRNA) targeting *pcnt*, two complementary oligonucleotides (sequences in *Supplementary file 2*) corresponding to a target sequence in the exon 2 of *pcnt* were annealed and cloned into pT7-gRNA plasmid to generate pT7-pcnt-gRNA. *pcnt* gRNA was generated by in vitro transcription using the MEGAshortscript T7 kit (AM1354, Thermo Fisher Scientific, Waltham, MA) with BamHI-linearized pT7-pcnt-gRNA as the template. Capped, zebrafish codon-optimized, double nuclear localization signal (nls)-tagged Cas9 RNA, *nls-zCas9-nls*, was synthesized by in vitro transcription using the mMESSAGE mMACHINE T3 kit (AM1348, Thermo Fisher Scientific) with XbaI-linearized pT3TS-nls-zCas9-nls plasmid as the template.

Microinjection of the mix of *pcnt* gRNA and *nls-zCas9-nls* RNA into zebrafish embryos (F0) was performed as described (*Jao et al., 2012*). Pipettes were pulled on a micropipette puller (Model P-97, Sutter Instruments, Novato, CA). Injections were performed with an air injection apparatus (Pneumatic MPPI-2 Pressure Injector, Eugene, OR). Injected volume was calibrated with a microruler (typically ~1 nl of injection mix was injected per embryo). Injected F0 embryos were raised and crossed with wild-type zebrafish to generate F1 offspring. Mutations in F1 offspring were screened by PCR amplifying the target region (primer sequences are in *Supplementary file 3*), followed by 7.5% acrylamide gel electrophoresis to detect heteroduplexes and sequencing. Two frameshift mutant alleles of *pcnt*, *pcnt*[tup2] and *pcnt*[tup5], were used in this study (*Figure 1—figure supplement 1*). Maternal-zygotic *pcnt* mutant embryos were generated by intercrosses of homozygous *pcnt*[tup2] or *pcnt*[tup5] fish.

## Inhibition of protein synthesis of zebrafish early embryos

To inhibit protein synthesis in blastula-stage zebrafish embryos, one-cell stage embryos from wild-type NHGRI-1 intercrosses were injected with ~1 nl of Injection Buffer alone (10 mM HEPES, pH 7.0, 60 mM KCl, 3 mM $MgCl_2$, and 0.05% phenol red) or with 300 µM puromycin in Injection Buffer. The embryos were fixed and analyzed after they developed to the two-cell stage.

## Cell culture

HeLa cells (ATCC CCL-2, a gift from Susan Wente, Vanderbilt University, Nashville, TN, or a HeLa cell line stably expressing scFv-sfGFP-GB1 and NLS-tdPCP-tdTomato, a gift from Xiaowei Zhuang, Howard Hughes Medical Institute, Harvard University, Cambridge, MA; *Wang et al., 2016*) and RPE-1 cells (a gift from Irina Kaverina, Vanderbilt University) or Centrin-GFP RPE-1 cells (a gift from Alexey Khodjakov, Wadsworth Center, New York State Department of Health, Rensselaer Polytechnic Institute, Albany, NY; *Uetake et al., 2007*) were maintained in Dulbecco's Modification of Eagles Medium (10–017-CV, Corning, Tewksbury, MA) and Dulbecco's Modification of Eagles Medium/ Ham's F-12 50/50 Mix (10–092-CV, Corning), respectively. All cell lines were supplemented with 10% fetal bovine serum (FBS) (12303C, lot no. 13G114, Sigma-Aldrich, St. Louis, MO), 1x Penicillin Streptomycin (30–002 CI, Corning), and maintained in a humidified incubator with 5% $CO_2$ at 37°C. To inhibit cytoplasmic dynein activities, the cells were treated with 50 µM ciliobrevin D for 1 hr 25 min at 37°C.

Cell lines used in this study were not further authenticated after obtaining from the sources. All cell lines were tested negative for mycoplasma using a PCR-based testing with the Universal Mycoplasma Detection Kit (30–1012K, ATCC, Manassas, VA). None of the cell lines used in this study were included in the list of commonly misidentified cell lines maintained by International Cell Line Authentication Committee.

## Cell synchronization

### Early M phase

Cells were synchronized by either double thymidine block using 2 mM thymidine (*Jackman and O'Connor, 2001*) or by the RO-3306 protocol using 6 µM RO-3306 (*Vassilev et al., 2006*). For HeLa, RPE-1, and Centrin-GFP RPE-1 cells, prophase and prometaphase cells were enriched in the cell population ~8 hr after the second release in the double thymidine block protocol, or 20–25 min after releasing cells from an 18 hr RO-3306 treatment.

## G1 phase

Cells were incubated with 6 µM RO-3306 for 18 hr, washed out, and incubated in fresh media with 10% FBS for 30 min. Mitotic cells were collected after two firm slaps on the plate and were plated again to circular coverslips. The cells were grown for 6 hr; at this time, almost all cells are in G1 phase (i.e. two centrin dots per cell).

## RNA in situ hybridization in zebrafish

In situ hybridizations of zebrafish embryos were performed as described (*Thisse and Thisse, 2008*). In brief, the DNA templates for making in situ RNA probes were first generated by RT-PCR using Tri-zol extracted total RNA from wild-type zebrafish oocytes as the template and gene-specific primers with T7 or T3 promoter sequence (sequences in *Supplementary file 3*). Digoxygenin-labeled anti-sense RNA probes were then generated by in vitro transcription and purified by ethanol precipitation (sequences in *Supplementary file 1*). Blastula-stage embryos were fixed in 4% paraformaldehyde in 1x PBS with 0.1% Tween 20 (1x PBS Tw) overnight at 4°C, manually dechorionated, and pre-hybridized in hybridization media (65% formamide, 5x SSC, 0.1% Tween-20, 50 µg/ml heparin, 500 µg/ml Type X tRNA, 9.2 mM citric acid) for 2–5 hr at 70°C, and hybridized for ~18 hr with hybridization media containing diluted antisense probe at 70°C. After hybridization, embryos were successively washed with hybridization media, 2x SSC with 65% formamide, and 0.2x SSC at 70°C, and finally washed with 1x PBS Tw at 25°C. Embryos were then incubated for 3–4 hr with blocking solution (2% sheep serum, 2 mg/ml BSA, 0.1% Tween-20 in 1x PBS) at 25°C, and incubated ~18 hr with blocking buffer containing anti-digoxigenin-alkaline phosphatase antibody (1:5000 dilution) at 4°C. Embryos were washed successively with 1x PBS Tw and AP Buffer (100 mM Tris, pH 9.5, 100 mM NaCl, 5 mM MgCl$_2$, 0.1% Tween-20) before staining with the NBT/BCIP substrates (11383213001/11383221001, Roche Diagnostics) in AP Buffer.

For combined RNA in situ hybridization and immunofluorescence to label both the RNA and centrosomes in zebrafish embryos, the RNA in situ hybridization process was performed as described above until the antibody labeling step: The embryos were incubated for ~18 hr with blocking solution (2% sheep serum, 2 mg/ml BSA, 0.1% Tween-20 in 1x PBS) containing anti-digoxigenin-peroxidase (1:500 dilution), anti-γ-tubulin (1:1000 dilution), and/or anti-phospho-Histone H3 (1:500 dilution) antibodies at 4°C. Embryos were washed successively with 1x PBS Tw and then incubated for ~18 hr with blocking solution containing Alexa Fluor 568 anti-mouse secondary antibody (1:500 dilution). After secondary antibody incubation, embryos were washed successively with 1x PBS and borate buffer (100 mM boric acid, 37.5 mM NaCl, pH 8.5) with 0.1% Tween-20. The RNA was visualized after tyramide amplification reaction by incubating embryos for 25 min in tyramide reaction buffer (100 mM boric acid, 37.5 mM NaCl, 2% dextran sulfate, 0.1% Tween-20, 0.003% H$_2$O$_2$, 0.15 mg/ml 4-iodophenol) containing diluted Alexa Fluor 488 tyramide at room temperature. The reaction was stopped by incubating embryos for 10 min with 100 mM glycine, pH 2.0 at room temperature, followed by successive washes with 1x PBS Tw.

## Fluorescent in situ hybridization with tyramide signal amplification (TSA) in cultured human cells

In brief, the DNA templates for making in situ RNA probes were first generated by RT-PCR using Tri-zol extracted total RNA from human 293 T cells as the template and gene-specific primers with T7 or T3 promoter sequence (sequences in *Supplementary file 3*). Digoxygenin-labeled antisense RNA probes were then generated by in vitro transcription and purified by ethanol precipitation (sequences in *Supplementary file 1*). Cells were fixed for ~18 hr with 70% ethanol at 4°C, rehydrated with 2x SSC (0.3 M NaCl, 30 mM trisodium citrate, pH 7.0) containing 65% formamide at room temperature, pre-hybridized for 1 hr with hybridization media (65% formamide, 5x SSC, 0.1% Tween-20, 50 µg/ml heparin, 500 µg/ml Type X tRNA, 9.2 mM citric acid) at 70°C, and hybridized for ~18 hr with hybridization media containing diluted antisense probes at 70°C. Cells were then successively washed with hybridization media, 2x SSC with 65% formamide, and 0.2x SSC at 70°C, and finally washed with 1x PBS at room temperature. For tyramide signal amplification, cells were washed with 1x PBS, incubated for 20 min with 100 mM glycine, pH 2.0, and washed with 1x PBS at room temperature. Cells were then incubated for 1 hr with blocking buffer (2% sheep serum, 2 mg/ml BSA, 0.1% Tween-20 in 1x PBS) at room temperature, and incubated ~18 hr with blocking buffer

containing anti-digoxigenin-peroxidase antibody (1:500 dilution) at 4°C. Cells were washed successively with 1x PBS, borate buffer (100 mM boric acid, 37.5 mM NaCl, pH 8.5) with 0.1% Tween-20, and incubated for 5 min in tyramide reaction buffer (100 mM boric acid, 37.5 mM NaCl, 2% dextran sulfate, 0.1% Tween-20, 0.003% $H_2O_2$, 0.15 mg/ml 4-iodophenol) containing diluted Alexa Fluor tyramide at room temperature. Cells were washed successively with 1x quenching buffer (10 mM sodium ascorbate, 10 mM sodium azide, 5 mM Trolox in 1x PBS) and 1x PBS at room temperature. Coverslips were mounted using ProLong Antifade media (P7481, Life Technologies).

## Sequential immunofluorescence (IF) and RNA single molecule fluorescent in situ hybridization (smFISH)

Sequential IF and smFISH were performed according to the manufacturer's protocol (LGC Biosearch Technologies, Petaluma, CA) with the following modifications: IF was performed first. Cells were fixed for 10 min in 4% paraformaldehyde in 1x PBS, washed twice with 1x PBS, and permeabilized with 0.1% Triton X-100 in 1x PBS for 5 min at room temperature. Cells were washed once with 1x PBS and incubated with 70 µl of diluted primary antibody in 1x PBS for 1 hr at room temperature. Cells were washed three times with 1x PBS and incubated with 70 µl of diluted secondary antibody in 1x PBS for 1 hr at room temperature. Cells were washed three times with 1x PBS and post-fixed for 10 min in 3.7% formaldehyde in 1x PBS at room temperature. For the smFISH process, cells were washed with Wash Buffer A, incubated with 67 µl of Hybridization Buffer containing 125 nM DNA probes labeled with Quasar 670 (sequences in *Supplementary file 1*) for 6 hr at 37°C. Cells were then incubated with Wash Buffer A for 30 min at 37°C, Wash Buffer A containing 0.05 µg/ml DAPI for 30 min at 37°C, and Wash Buffer B for 3 min at room temperature. Coverslips were mounted using ProLong Antifade media (Life Technologies) and sealed with clear nail polish before imaging.

## Immunofluorescence

Cells were fixed for 10 min in 4% paraformaldehyde in 1x PBS, washed twice with 1x PBS, and permeabilized with 0.5% Triton X-100 in 1x PBS for 5 min at room temperature. Cells were incubated with blocking solution (2% goat serum, 0.1% Triton X-100, and 10 mg/ml of bovine serum albumin in 1x PBS) for 1 hr at room temperature, incubated with blocking solution containing diluted primary antibody for 1 hr at room temperature. Cells were washed three times with 1x PBS and incubated with blocking solution containing diluted secondary antibody for 1 hr at room temperature. Cells were washed with 1x PBS and nuclei were counterstained with 0.05 µg/ml of DAPI in 1x PBS for 20 min at room temperature before mounting.

## EdU labeling

S phase cells were detected by using the Click-iT EdU Imaging Kit (Life Technologies) according to the manufacturer's instruction. In brief, Centrin-GFP RPE-1 cells were grown on 12-mm acid-washed coverslips and pulse labeled with 10 µM 5-ethynyl-2′-deoxyuridine (EdU) for 30 min at 37°C. The cells were then fixed for 10 min with 4% paraformaldehyde in 1x PBS at room temperature, washed twice with 1x PBS, and permeabilized for 20 min with 0.5% Triton X-100 in 1x PBS. Cells were then washed twice with 1x PBS and incubated with a Click-iT cocktail mixture containing Alexa Fluor 488 or 594 azide for 30 min in the dark at room temperature.

## Microscopy

Embryos subjected to in situ hybridization were mounted in a 35-mm glass-bottom dish (P35G-1.5–10 C, MatTek, Ashland, MA) in 0.8% low melting point agarose and imaged using a stereo microscope (M165 FC, Leica, Wetzlar, Germany) with a Leica DFC7000 T digital camera.

Confocal microscopy was performed using either a Leica TCS SP8 laser-scanning confocal microscope system with 63x/1.40 or 100x/1.40 oil HC PL APO CS2 oil-immersion objectives and HyD detectors in resonant scanning mode, or a spinning disk confocal microscope system (Dragonfly, Andor Technology, Belfast, UK) housed within a wrap-around incubator (Okolab, Pozzuoli, Italy) with Leica 63x/1.40 or 100x/1.40 HC PL APO objectives and an iXon Ultra 888 EMCCD camera for smFISH and live cell imaging (Andor Technology). Deconvolution was performed using either the Huygens Professional (Scientific Volume Imaging b.v., Hilversum, Netherlands) (for images captured on Leica SP8) or the Fusion software (Andor Technology) (for images captured on Andor Dragonfly).

## Quantification of smFISH data and PCNT levels at centrosomes

To quantify the RNA distribution within the cell in 3D voxels, we used Imaris software (Bitplane, Belfast, UK) to fit the protein signal as surfaces and the mRNA signal as spots of different sizes in deconvolved images of each confocal z-stack. The intensity of the mRNA signal in each spot is assumed to be proportional to the amount of mRNA in each spot and is used in lieu of mRNA units. The outline of the cell was obtained either from a transmitted light image or from the background in the pre-deconvolved image and was used to restrict fitting of both mRNA and protein signals to the cell of interest. The distance from each mRNA spot to each centrosome's center of mass was calculated and the mRNA signal was 'assigned' to the closest centrosome. The mRNA spots were binned by distance to the centrosome and the intensities of the spots in each bin were added as a measure of the amount of mRNA at that distance. This was calculated for each cell and then averaged over all the cells for each condition. Thus, the graphs show average mRNA as a function of distance (binned in 0.5 μm intervals).

To quantify PCNT intensities at the centrosome, we put the surfaces of the anti-PCNT signals fit on the deconvolved images over the original images and used the statistics function in Imaris (Bitplane) to obtain the intensity sum of the original images within the fit volume.

## Live translation assay (SunTag/PP7 system)

A HeLa cell line stably expressing scFv-sfGFP-GB1 and NLS-tdPCP-tdTomato was transfected with the SunTag/PP7 reporter plasmid pEF-24xV4-ODC-24xPP7 (*Wang et al., 2016*) using Lipofectamine 3000 transfection reagent (Life Technologies) according to the manufacturer's instruction. 12–18 hr after transfection, the medium was changed to 10% FBS/DMEM without phenol red before imaging.

## Statistical analysis

Statistical analysis was performed using the GraphPad Prism 7. Each exact $n$ value is indicated in the corresponding figure or figure legend. Significance was assessed by performing an unpaired two-sided Student's t-test, as indicated in individual figures. The experiments were not randomized. The investigators were not blinded to allocation during experiments and outcome assessment.

## Acknowledgements

We thank Susan Wente for the HeLa cell line; Irina Kaverina for the RPE-1 cell line; Alexey Khodjakov for the Centrin-GFP RPE-1 stable cell line; Xiaowei Zhuang for the SunTag/PP7 reporter plasmid pEF-24xV4-ODC-24xPP7, and the HeLa cell line stably expressing scFv-sfGFP-GB1 and NLS-tdPCP-tdTomato; Dena Leerberg and Bruce Draper for technical help on fluorescent in situ hybridization in zebrafish; Stephen (Evan) Brahms, Marvin Orellana, Hashim Shaikh, Janice Tam, and Alan Zhong for technical help on zebrafish and cell culture work; Tom Glaser, Henry Ho, Frank McNally, Richard Tucker, and Mark Winey for critical reading of the manuscript; Emily Jao for help on digital illustrations. Experiments were performed in part through the use of UC Davis Health Sciences District Advanced Imaging Facility.

## Additional information

### Funding

| Funder | Grant reference number | Author |
| --- | --- | --- |
| University of California, Davis | New Faculty Startup Funds | Li-En Jao |

The funders had no role in study design, data collection and interpretation, or the decision to submit the work for publication.

### Author contributions

Guadalupe Sepulveda, Mark Antkowiak, Data curation, Formal analysis, Investigation, Methodology, Writing—original draft, Writing—review and editing; Ingrid Brust-Mascher, Resources, Data curation, Software, Formal analysis, Validation, Investigation, Visualization, Methodology, Writing—original

draft, Writing—review and editing; Karan Mahe, Tingyoung Ou, Noemi M Castro, Lana N Christensen, Lee Cheung, Data curation, Formal analysis, Investigation; Xueer Jiang, Data curation, Software, Formal analysis, Investigation; Daniel Yoon, Investigation; Bo Huang, Data curation, Formal analysis, Methodology; Li-En Jao, Conceptualization, Resources, Data curation, Formal analysis, Supervision, Funding acquisition, Validation, Investigation, Visualization, Methodology, Writing—original draft, Project administration, Writing—review and editing

### Author ORCIDs
Li-En Jao (iD) http://orcid.org/0000-0001-5925-883X

### Ethics
Animal experimentation: This study was performed in strict accordance with the recommendations in the Guide for the Care and Use of Laboratory Animals of the National Institutes of Health. All of the animals were handled according to approved institutional animal care and use committee (IACUC) protocols (#20169) of the University of California, Davis.

### Decision letter and Author response
Decision letter https://doi.org/10.7554/eLife.34959.034
Author response https://doi.org/10.7554/eLife.34959.035

## Additional files

### Supplementary files
• Supplementary file 1. Sequences of antisense probes used in RNA in situ hybridization in zebrafish and cultured cells.
DOI: https://doi.org/10.7554/eLife.34959.029

• Supplementary file 2. Sequence of the genomic target site of zebrafish *pcnt* and oligonucleotides for making the customized gRNA expression construct. The sense strand of the target site is shown.
DOI: https://doi.org/10.7554/eLife.34959.030

• Supplementary file 3. PCR primer sequences for amplifying the zebrafish *pcnt* CRISPR target region and generating antisense probes for in situ hybridization.
DOI: https://doi.org/10.7554/eLife.34959.031

• Transparent reporting form
DOI: https://doi.org/10.7554/eLife.34959.032

### Data availability
All data generated or analyzed during this study are included in the manuscript and supporting files. Source data files have been provided for Figure 2B, Figure 3C, Figure 3—figure supplement 2, Figure 4A, Figure 4D, Figure 4—figure supplement 2, Figure 4—figure supplement 3B, Figures 5A-5C, Figure 6, and Figure 6—figure supplement 1.

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
