## [Decision Letter]

Thank you for submitting your article "Co-translational protein targeting facilitates centrosomal recruitment of PCNT during centrosome maturation" for consideration by *eLife*. Your article has been favorably evaluated by Andrea Musacchio (Senior Editor) and three reviewers, one of whom, Yukiko M Yamashita (Reviewer #1), is a member of our Board of Reviewing Editors. The following individual involved in review of your submission has agreed to reveal their identity: Meng-Fu Bryan Tsou (Reviewer #3).

The reviewers have discussed the reviews with one another and the Reviewing Editor has drafted this decision to help you prepare a revised submission.

The manuscript by Sepulveda et al. provides convincing evidence for the discovery that some large, PCM-associated centrosomal proteins like PCNT can be targeted/transported to the centrosome when they are still being translated on the polysome as partially formed polypeptides, a process called "co-translational protein targeting". They showed that targeting of the PCNT-associated polysome to the centrosome is dependent on dynein and MT networks, and is required for efficient centrosome maturation during mitosis. The authors also identified a similar mechanism by which the ASPM-associated polysome is targeted to the spindle pole, where ASPM is known to facilitate proper spindle assembly. The story essentially describes a new category of posttranscriptional regulation of centrosomal genes, and is thus significant/interesting/novel.

The only major concern that the reviewers unanimously agreed was the fact that the most critical data to support their central hypothesis shown in Figure 3 are not strong enough in its current form. It relies on colocalization of PCNT mRNA together with N-ter PNCT antibody but not C-ter PCNT antibody. Because the data in this figure are of fundamental importance for this study, the reviewers would like the data in this figure and the conclusion drawn from it to be solid at the highest level of rigor. Around this issue, the reviewers have several suggestions as summarized below. The authors do not necessarily have to do all of them, or they can employ any other methods of their choice, but the collective sum of the revision on this issue must be rigorously supporting their conclusion that PCNT mRNA around the centrosome contains nascent PCNT polypeptides that are undergoing translation.

1) The results in Figure 3 relies on just one N-ter antibody and one C-ter antibody. The concern was raised regarding the specificity of this antibody. In particular, the presentation only focuses on the vicinity of the centrosome only, and the reviewers/readers cannot judge the degree of its background. It is possible that showing low magnification image together with magnified insets (similar to Figure 3—figure supplement 1B, C) might sufficiently address the concern. Alternatively, additional N-ter, C-ter antibodies, or N-ter/C-ter tagging etc. (if available/feasible) would address this concern.

2) Additionally, it would help if the authors can colocalize PCNT mRNA/nascent polypeptide with ribosomes (e.g. antibody staining against ribosome subunits).

3) The authors could expand the panel C and do a cyclohexamide (CHX) treatment. CHX should lock the ribosomes onto the transcript, and the predicted result would be higher coincidence of RNA and N-term protein signals, likely higher than the 10% seen in controls.

4) Any other ways to make the data in Figure 3 robust will be acceptable.

---

## [Author Response]

[…] The only major concern that the reviewers unanimously agreed was the fact that the most critical data to support their central hypothesis shown in Figure 3 are not strong enough in its current form. It relies on colocalization of PCNT mRNA together with N-ter PNCT antibody but not C-ter PCNT antibody. Because the data in this figure are of fundamental importance for this study, the reviewers would like the data in this figure and the conclusion drawn from it to be solid at the highest level of rigor. Around this issue, the reviewers have several suggestions as summarized below. The authors do not necessarily have to do all of them, or they can employ any other methods of their choice, but the collective sum of the revision on this issue must be rigorously supporting their conclusion that PCNT mRNA around the centrosome contains nascent PCNT polypeptides that are undergoing translation.1) The results in Figure 3 relies on just one N-ter antibody and one C-ter antibody. The concern was raised regarding the specificity of this antibody. In particular, the presentation only focuses on the vicinity of the centrosome only, and the reviewers/readers cannot judge the degree of its background. It is possible that showing low magnification image together with magnified insets (similar to Figure 3—figure supplement 1B, C) might sufficiently address the concern. Alternatively, additional N-ter, C-ter antibodies, or N-ter/C-ter tagging etc. (if available/feasible) would address this concern.

In the updated Figure 3, we have included the low magnification images together with magnified insets to address the concern of antibody specificity. For better visualization, individual channels of the low magnification images are shown in monochrome, along with the merged images shown in color.

2) Additionally, it would help if the authors can colocalize PCNT mRNA/nascent polypeptide with ribosomes (e.g. antibody staining against ribosome subunits).

We have performed double immunostaining against PCNT N-terminus and ribosomal protein S6 along with *PCNT* smFISH in both HeLa and RPE-1 cells. By assessing colocalization of these three signals in thin optical sections, we found that during early mitosis, *PCNT* mRNA molecules positive for anti-PCNT N-term signals near the centrosome were often also positive for anti-ribosomal protein S6 signals. These data further support the conclusion that a portion of *PCNT* mRNA near the centrosome is undergoing active translation during early mitosis. These new data are presented in Figure 3—figure supplement 3.

3) The authors could expand the panel C and do a cyclohexamide (CHX) treatment. CHX should lock the ribosomes onto the transcript, and the predicted result would be higher coincidence of RNA and N-term protein signals, likely higher than the 10% seen in controls.

Because antibody detection is not at the single molecule level, detecting nascent PCNT polypeptides by immunostaining on a single *PCNT* mRNA molecule requires multiple ribosomes translating a single mRNA so that multiple copies of nascent PCNT polypeptides can be detected by anti-PCNT N-term antibody. Therefore, in the initial phase of translation, a given *PCNT* mRNA will be negative for anti-PCNT N-term signals (even if it is already undergoing active translation). As additional ribosomes initiate translation on the same *PCNT* mRNA and additional nascent PCNT polypeptides are made in the polysome, this *PCNT* mRNA would then become visible by anti-PCNT N-term staining. When no more ribosome initiates translation on the same *PCNT* mRNA, the ability of detecting anti-PCNT N-term signals on the *PCNT* mRNA would gradually decrease as the existing ribosomes are running off from the mRNA.

With this dynamic process in mind, adding CHX or emetine, which freezes ribosomes and halts translation, would have two effects on the overall coincidence of *PCNT* mRNA and anti-PCNT N-term signals. First, it would prevent the existing ribosomes from running off from the *PCNT* mRNA, potentially leading to higher coincidence of detecting anti-PCNT N-term signals on a *PCNT* mRNA (i.e. the scenario the reviewers referred to). However, at the same time, CHX or emetine would also have a second effect: the *PCNT* mRNA molecules that are at the early stage of translation would never have the chance to be detected by anti-PCNT N-term antibody because not enough copies of nascent PCNT polypeptides could have been made. Therefore, the effect of CHX or emetine on the overall coincidence of *PCNT* mRNA and anti-PCNT N-term signals would depend on the balance between preserving the “old” polypeptides and failing to make the “new” polypeptides that would have been made should CHX or emetine not be added.

Based on this reasoning, we predicted that adding CHX or emetine could increase, decrease, or have no effect on the overall percentage of PCNT N-term-positive *PCNT* mRNA and that the length of translation inhibition could potentially shift the balance toward either end. This prediction is in fact correct: We performed additional CHX experiments and quantified the results at the single cell level. We found that the short “freeze” of polysomes (400 µM CHX for 2 minutes at 37°C) resulted in a slightly higher coincidence of anti-PCNT N-term and *PCNT* mRNA signals as compared to the control (*p*=0.0449, n=125 and 120 for control and CHX samples, respectively, from two independent biological replicates). However, when quantifying our existing emetine data, which involved a longer, 30-minute freeze of translation elongation, we observed a significant lower coincidence of anti-PCNT N-term and *PCNT* mRNA signals compared to the control (*p*=0.0008, n=48 and 45 for control and emetine samples, respectively, from three independent biological replicates).

Unlike the counteracting effects of CHX and emetine in this assay, the puromycin treatment would always have a negative effect on the overall coincidence of PCNT N-term and *PCNT* mRNA signals. For this reason, we believe that applying puromycin and assessing whether the overall coincidence of anti-PCNT N-term and *PCNT* mRNA signals is altered would provide a more reliable readout than using CHX or emetine to determine if *PCNT* mRNA is undergoing active translation. We have consistently observed that the acute puromycin treatment (37°C for 2 minutes) resulted in a significant loss of PCNT N-term signals on *PCNT* mRNA near the centrosome (Figure 3 and data not shown). In addition, this short puromycin treatment was sufficient to shift the *PCNT* mRNA distribution away from the centrosome (Figure 4). Together, these two results strongly suggest that *PCNT* mRNA is undergoing active translation near centrosomes during early mitosis.

4) Any other ways to make the data in Figure 3 robust will be acceptable.

We did not pursue alternative approaches other than the experiments described above.